



# A machine-learning reconstruction of sea surface $p$CO₂ in the North American Atlantic Coastal Ocean Margin from 1993 to 2021

Zelun Wu[1,2], Wenfang Lu[3*], Alizée Roobaert[4], Luping Song[5], Xiao-Hai Yan[2], and Wei-Jun Cai[2]

[1] State Key Laboratory of Marine Environmental Science & College of Ocean and Earth Science, Xiamen University, Xiamen,
Fujian, China, 361102
[2] School of Marine Science and Policy, University of Delaware, Newark, Delaware, USA, 19716
[3] School of Marine Sciences, Sun Yat-sen University, & Southern Marine Science and Engineering Guangdong Laboratory
(Zhuhai), Zhuhai, Guangdong, China, 519082
[4] Flanders Marine Institute (VLIZ), Jacobsenstraat 1, Ostend, Belgium, 8400
[5] School of Marine Science and Technology, Zhejiang Ocean University, Zhoushan, Zhejiang, China, 316022

*Correspondence to*: Wenfang Lu (luwf6@sysu.edu.cn)

**Abstract.** Insufficient spatiotemporal coverage of partial pressure of CO₂ ($p$CO₂) observations has hindered precise studies of
the coastal carbon cycle along the North American Atlantic Coastal Ocean Margin (NAACOM). Earlier $p$CO₂-products have
encountered difficulties in accurately capturing the heterogeneity of regional variations and decadal trends of $p$CO₂ in the
NAACOM. This study developed a regional reconstructed $p$CO₂-product for the NAACOM (Reconstructed Coastal
Acidification Database-$p$CO₂, or ReCAD-NAACOM-$p$CO₂) using a two-step approach combining random forest regression
and linear regression. The product provides monthly $p$CO₂ data at 0.25° spatial resolution from 1993 to 2021, enabling
investigation of regional spatial differences, seasonal cycles, and decadal changes in $p$CO₂. The observation-based
reconstruction was trained using Surface Ocean CO₂ Atlas (SOCAT) observations as ground-truth values, with various
satellite-derived and reanalysis environmental variables known to control sea surface $p$CO₂ as model inputs. The product shows
high accuracy during the model training, validation, and independent test phases, demonstrating robustness and capability to
accurately reconstruct $p$CO₂ in regions or periods lacking direct observational data in the NAACOM. Compared with all the
observation samples from SOCAT, the $p$CO₂-product yields a determination coefficient of 0.83, a root-mean-square error of
18.64 µatm, and an accumulative uncertainty of 23.83 µatm. The ReCAD-NAACOM-$p$CO₂ product demonstrates its capability
to resolve seasonal cycles, regional-scale variations, and decadal linear trends of $p$CO₂ along the NAACOM. This new product
provides reliable $p$CO₂ data for more precise studies of coastal carbon dynamics in the NAACOM region. The dataset is
publicly accessible at https://doi.org/10.5281/zenodo.11500974 (Wu et al., 2024a) and will be updated regularly.



## 1 Introduction

The coastal ocean, despite covering only 8.4% ($30.4 \times 10^6$ km$^2$) of the global ocean surface area (Chen et al., 2013; Dai et al.,
2022), plays a disproportionately significant role in the global carbon budget, accounting for approximately 10.9% of the
global ocean $CO_2$ uptake from the atmosphere (0.25 of 2.3 Pg C yr$^{-1}$) on the global average (Dai et al., 2022; Friedlingstein et
al., 2023). However, accurately quantifying the $CO_2$ uptake in specific coastal regions only based on observations is
challenging due to the scarcity of sea surface partial pressure of $CO_2$ ($p$CO$_2$) data. Moreover, in coastal regions, sea surface
$p$CO$_2$ is highly variable due to the influence of various physical and biogeochemical processes, such as riverine input,
upwelling, tidal mixing, and large-scale circulations (Laruelle et al., 2018; Roobaert et al., 2024b). Accurate and
comprehensive $p$CO$_2$ data are necessary to quantify coastal $CO_2$ uptake and assess the impact of climate change on coastal
ocean ecosystems.

This study focuses on the North American Atlantic Coastal Ocean Margin (NAACOM, **Fig. 1**). The entire region is defined as
the area within 400 km of the coastline and divided into six sub-regions based on their geographic location following Fennel
et al. (2019), including the Gulf of Mexico (GoMx), South Atlantic Bight (SAB), Mid-Atlantic Bight (MAB), Gulf of Maine
(GoMe), Scotian Shelf (SS), and Gulf of St. Lawrence & Grand Banks (GStL&GB). The carbonate system in the NAACOM
is influenced by large-scale circulations (**Fig. 1**), including the Gulf Stream and Labrador Current, as well as local processes
like river discharge, export from marshes, and upwellings dynamics (Cai et al., 2020; Fennel et al., 2019; Wang et al., 2013).
These complex physical and biogeochemical processes contribute to substantial spatial and temporal heterogeneity in sea
surface $p$CO$_2$ across the NAACOM (Cai et al., 2020). Elucidating the driving mechanisms of spatial and temporal $p$CO$_2$
variations necessitates extensive data coverage in time and space in this region. Over the past two decades, coastal field
investigation efforts in this region have substantially increased through programs like the East Coast Ocean Acidification
(ECOA) and Gulf of Mexico Ecosystems and Carbon Cruise (GOMECC) (Cai et al., 2020; Wang et al., 2013; Wanninkhof et
al., 2015). Data from these cruises, combined with underway measurements and buoy observations, are quality-controlled and
compiled in the Surface Ocean $CO_2$ Atlas (SOCAT) database (Bakker et al., 2016), substantially advancing our understanding
of coastal inorganic carbon chemistry along the NAACOM (Cai et al., 2020).

Despite significant progress in observational efforts, the spatial and temporal coverage of $p$CO$_2$ data remains limited in the
NAACOM, with observations encompassing only 2.9 % of grid cells during the period 1993-2021 (**Fig. 2**). Observations are
concentrated in the southern regions, with fewer samples available during winter. This data scarcity introduces substantial
uncertainty in the air-sea $CO_2$ exchange quantification and hinders a comprehensive understanding of coastal inorganic carbon
dynamics, particularly in areas north of Cape Cod where measurements are highly sparse (**Fig. 2**). For example, reported air-

sea $CO_2$ fluxes for the GoMe exhibit a wide range, spanning from -0.50 to +2.50 mol C m$^{-2}$ yr$^{-1}$, with conflicting reports characterizing it as a $CO_2$ source (Fennel & Wilkin, 2009; Vandemark et al., 2011), $CO_2$ neutral (Signorini et al., 2013), and $CO_2$ sink (Cahill et al., 2016; Rutherford et al., 2021), underscoring the need for improved $p$CO$_2$ data coverage.

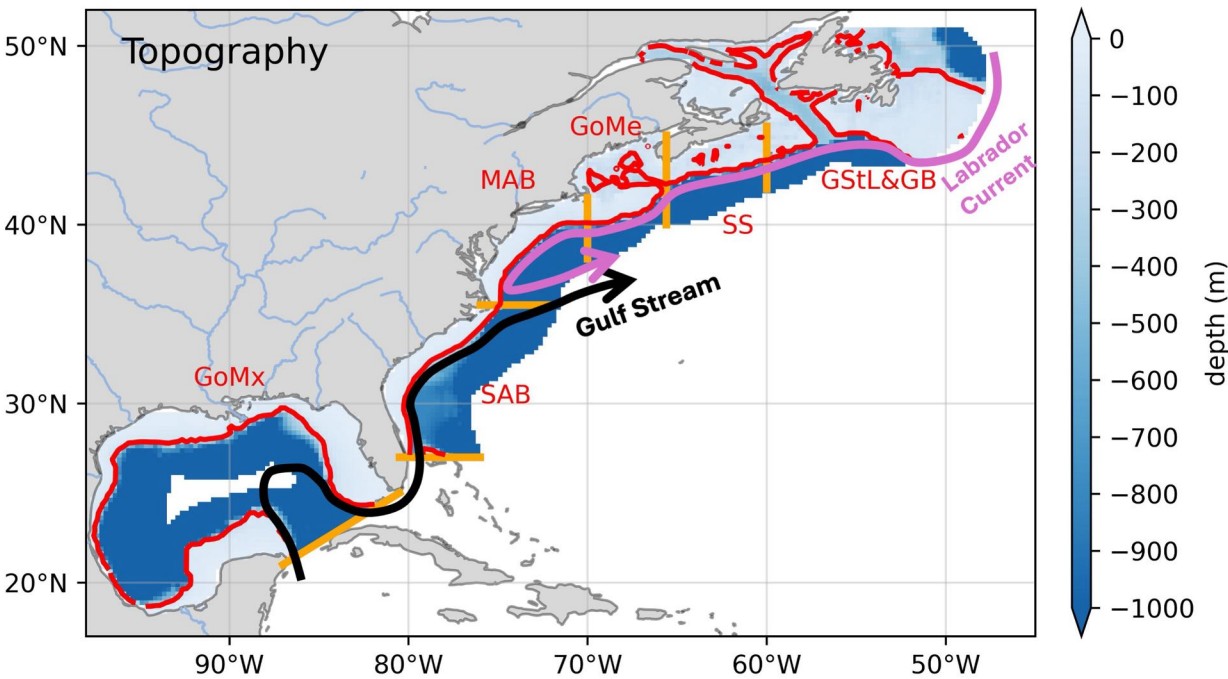

**Figure 1. Topography (in meters) and large-scale circulation along the North American Atlantic Coastal Ocean Margin (NAACOM).** The region is defined as coastal areas extending 400 km offshore. The thin red contour line is the 200 m isobath, which is roughly the location of the shelf break and a typical definition of continental shelf boundary**.** The Gulf Stream (thick black line with an arrow) flows northward along the east coast of the United States before veering eastward into the open Atlantic Ocean around Cape Hatteras. The Labrador Current (thick pink line with an arrow) flows southward along the east coast of Canada before meeting the Gulf Stream. In this study, the study region is divided into six sub-regions by the straight yellow lines, including the Gulf of Mexico (GoMx), South Atlantic Bight (SAB), Mid-Atlantic Bight (MAB), Gulf of Maine (GoMe), Scotian Shelf (SS), and Gulf of St. Lawrence and Grand Banks (GStL&GB) following Fennel et al. (2019).

Recently, various global-scale and regional reconstructed $p$CO$_2$-products with full coverage in time and space have been developed as essential supplements to observations. These products usually employed diverse algorithms and utilized environmental proxies from satellites and reanalysis products as model inputs and SOCAT observations as constraints to reconstruct the $p$CO$_2$ field with full temporal and spatial coverage. The development of those products has significantly



advanced our understanding of inorganic carbon chemistry and the ocean carbon cycle. For example, seven global $pCO_2$-products were used to evaluate the ocean $CO_2$ uptake in the Global Carbon Budget 2023 edition (Friedlingstein et al., 2023).

However, most of these products reconstruct $pCO_2$ in the open ocean, with coastal regions often being extrapolated or excluded. In contrast to the open ocean, where several global $pCO_2$-products have been developed this past decade, there are fewer $pCO_2$-products specifically designed for global coastal oceans. Currently, only one $pCO_2$-product has been developed specifically for the coastal ocean on a global scale (Laruelle et al., 2017; Roobaert et al., 2024a). This product was recently combined with an open ocean product to create a global reconstruction of the ocean $CO_2$ sink (Landschützer et al., 2020) and has since been

utilized to narrow the variability in global reconstructions (Fay et al., 2021). However, global products primarily aim to ensure high accuracy of parameters on a global average scale; they may not guarantee equivalent accuracy for spatiotemporal variations on the regional scale. In comparison, regional $pCO_2$-products have demonstrated superior capability in resolving detailed small-scale variations.

Within the NAACOM region, several area-specific $pCO_2$-products have been reconstructed, focusing on specific regions such

as the GoMx (e.g., Chen and Hu, 2019; Fu et al., 2020; Lohrenz and Cai, 2006) and the SAB and MAB (e.g., Wang et al., 2024; Xu et al., 2020). These regional and global $pCO_2$-products are valuable for validating model estimations (Roobaert et al., 2022; Ross et al., 2023). However, existing products often have limitations in spatial coverage, temporal resolution, or trend analysis capabilities. For instance, Chen and Hu (2019) provided a high-resolution (4 km) $pCO_2$-product for the GoMx, but this product faces challenges in capturing decadal changes in $pCO_2$ (Wu et al., 2024b). Conversely, Xu et al. (2020)

successfully captured decadal trends of $pCO_2$, but only as area-averaged $pCO_2$ time series for the SAB and MAB, lacking comprehensive spatial coverage. Signorini et al. (2013) reconstructed a product using multiple linear regression (MLR) covering the areas from SAB to SS, but it spans only 8 years (2003-2010). Despite these valuable efforts, there remains a lack of comprehensive data products that adequately capture regional variations, seasonal cycles, and decadal changes in $pCO_2$ simultaneously for the entire NAACOM.

This study aims to develop a regional $pCO_2$-product specifically designed for the NAACOM, encompassing coastal regions extending 400 km offshore from the GoMx to the GB (**Fig. 1**). We integrated random forest and linear regression methods with hydrological parameters from satellite observations and reanalysis data to generate a monthly reconstructed $pCO_2$-product at 0.25° spatial resolution spanning the period 1993 to 2021. The $pCO_2$-product, termed 'Reconstructed Coastal Acidification Database' or 'ReCAD-NAACOM-$pCO_2$', is specifically designed to resolve spatial variations, seasonal cycles, and decadal

changes of $pCO_2$ along the NAACOM.

The structure of this paper is as follows: Section 2 details the methodology used to reconstruct ReCAD-NAACOM-$p$CO$_2$ and describes the datasets employed. Section 3 evaluates the product's accuracy, performance, and applicability in resolving seasonal cycles, regional variations, and decadal trends of $p$CO$_2$. Sections 4 and 5 provide links to access the dataset and codes used for generating the dataset and figures presented in this study. The final section summarizes the conclusions. ReCAD-

NAACOM-$p$CO$_2$ demonstrates enhanced capability in resolving spatial variations and capturing the seasonal cycle and decadal trends of $p$CO$_2$ across different sub-regions along the NAACOM. This product offers improved insights into coastal carbon dynamics in this complex region, addressing the need for a comprehensive $p$CO$_2$ data in the NAACOM.

## 2 Data and methods

### 2.1 Ground-truth data from SOCAT

The ground-truth data for the training regression model were the seawater fugacity of CO$_2$ ($f$CO$_2$) measurement extracted from the SOCAT database (version 2023). $f$CO$_2$ represents the $p$CO$_2$ corrected for the non-ideal behavior of the gas in seawater, and both are commonly used in oceanographic studies. SOCAT compiles quality-controlled $f$CO$_2$ measurements from various platforms, including research vessels, commercial ships, and moorings (Bakker et al., 2016). This study used the monthly gridded SOCAT coastal product with a spatial resolution of 0.25° × 0.25° (but with data gaps). The gridded product

incorporated measurements with quality flags A, B (uncertainty of 2 µatm), C, and D (uncertainty of 5 µatm) (Bakker et al., 2016). Over the period 1993-2021, the SOCAT product encompassed 55,347 grid cells within our study area (**Fig. 2**), accounting for approximately 2.9% of the total grid cells in the NAACOM. Spatial analysis revealed lower sampling density in the areas north of Cape Cod (blue box in **Fig. 2**). The temporal distribution of samples exhibits a notable bias, with reduced collection during winter (**Fig. 2d**). Despite these spatial and temporal heterogeneities, the SOCAT observations provide

coverage across all sub-regions and seasons of the NAACOM (**Fig. 2**). This comprehensive, albeit sparse, coverage facilitates the reconstruction of the $f$CO$_2$ and $p$CO$_2$ field through interpolation and regression techniques.

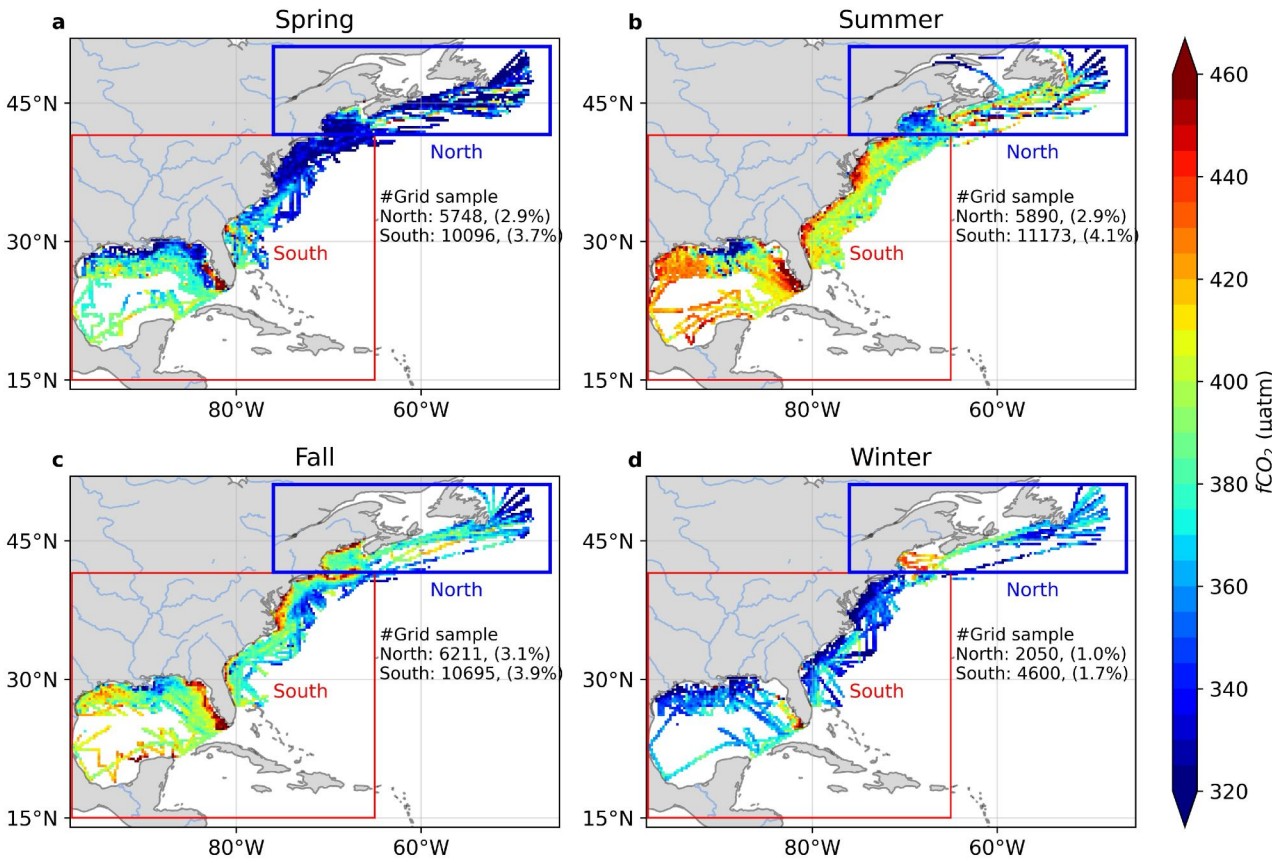

**Figure 2. Spatial distribution of sea surface _f_CO₂ observations from SOCAT database (version 2023) in the NAACOM across four seasons from 1993 to 2021.** Grid samples with data were counted by season: **(a)** Spring (March to May), **(b)** Summer (June to August), **(c)** Fall (September to November), and **(d)** Winter (December to February). The study region is divided into northern (blue box) and southern (red box) areas at approximately 41.5°N (Cape Cod). The number and percentage of grid samples are indicated for each region per season. Color scale represents _f_CO₂ values in µatm. Higher sampling density is evident in the southern area. Winter shows the lowest overall sampling coverage.





## 2.2 Model design

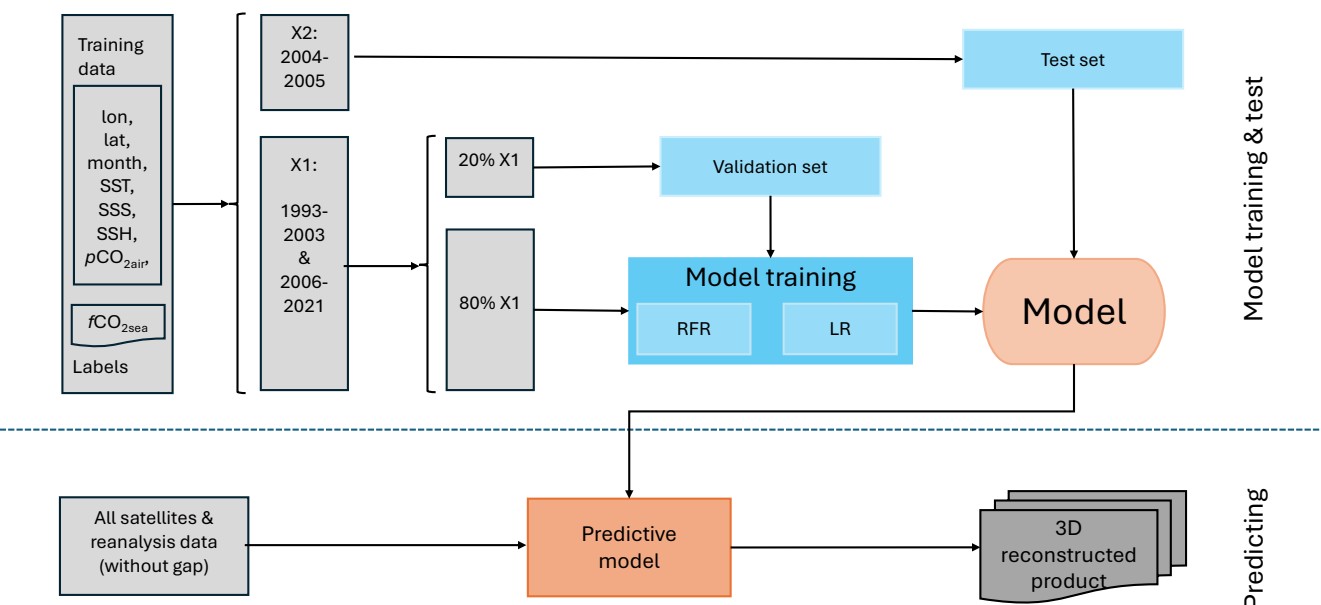


**Figure 3. A flowchart of the two-step machine learning regression model for generating the reconstructed $pCO_2$-product.** Grey boxes represent the input and output datasets, blue boxes illustrate the model training, validation testing, and independent test processes, and orange boxes represent the final trained model for predicting the reconstructed product. The training data, consisting of paired input variables (lon, lat, month, sea surface temperature (SST), sea surface salinity (SSS),

sea surface height (SSH), and atmospheric $pCO_2$ ($pCO_{2air}$) and corresponding sea surface $fCO_2$ ($fCO_{2sea}$) labels), is divided into two sets: X1 (1993-2003 and 2006-2021) and X2 (2004-2005). X1 is further randomly divided into subsets for model training set (80%) and validation set (20%). The predictive model combines a random forest regression (RFR) and a linear regression (LR) algorithm. The trained and validated regression model is then applied to all satellite and reanalysis data (without gaps) to generate the final 3D reconstructed $fCO_2$ product, which was finally converted to $pCO_2$.

The procedures of developing and reconstructing the $pCO_2$-product are illustrated in **Fig. 3**. Initially, the input variables and

sea surface $fCO_2$ data were matched to create a comprehensive dataset. To maintain consistency with the SOCAT database,

which reports sea water $CO_2$ concentrations as $fCO_2$, we adopted $fCO_2$ as the output variable in our model. The matched dataset

was then divided into two sets: X1, encompassing the periods 1993-2003 and 2006-2021, and X2, covering 2004-2005. Set

X1 was further randomly subdivided, with 80% allocated for model training and the remaining 20% for validation test. Set X2

served as an independent test set. The model training set (80% of X1) was used to develop a two-step RFR+LR regression





model. The RFR is designed to capture complex, nonlinear relationships between the input variables and the target variable (i.e., $fCO_2$), while the LR model is subsequently applied to mitigate potential systematic biases in RFR-derived $fCO_2$ values arise from spatiotemporal heterogeneities in the SOCAT observational dataset **(Fig. 2)**. RFR, an ensemble learning technique, combines multiple decision trees to produce more accurate and stable predictions (Breiman, 2001; Lu et al., 2019). Each

decision tree in the RFR is trained on a randomly selected subset of the input data, with the final prediction derived from the average output of all trees. This approach mitigates overfitting and enhances the model's generalization performance, making it particularly suitable for large datasets with complex, nonlinear variable relationships. The RFR model was trained using 10-fold cross-validation, with optimized hyperparameters including a minimum leaf size of 1, bagging method for ensemble aggregation, and 300 learning cycles after tuning. After RFR model training, an LR model was applied to the RFR-estimated

$fCO_2$ ($fCO_{2est}$) output to make sure the RFR model is not systematically biased:

$$fCO_{2obs} = a \times fCO_{2est} + b + \varepsilon \tag{1}$$

where $fCO_{2obs}$ is the observed $fCO_2$ from SOCAT, a is the linear regression coefficient, b is the intercept, and $\varepsilon$ is the residual that the linear model cannot resolve. This additional step was implemented to mitigate potential systematic bias in the RFR model that could arise from areas with higher sampling density, thereby ensuring a more balanced representation across the

entire study region. The calibration was applied to each grid cell individually. To increase the data pool for linear regression, samples within a 5 × 5 grid window in space (i.e., 1.25° × 1.25°) were aggregated for LR model development. As the available measurements could not cover every grid cell and were insufficient to produce continuous spatial maps of the calibration coefficients (i.e., a and b in Equation 1), we employed a locally interpolated regression strategy similar to Carter et al. (2018). Mathematically, given the spatial and temporal continuity of $fCO_{2est}$ and $fCO_{2obs}$, the coefficients a and b must also be

continuous in space and time. Therefore, we linearly interpolated the coefficients a and b across the NAACOM. The interpolated coefficients were subsequently used to adjust the RFR-derived $fCO_{2est}$.

The validation set (20% of X1) was used to evaluate the model's performance. This subset helps in tuning the model's hyperparameters and provides an unbiased evaluation of the model's performance, helping to prevent overfitting.

The independent test set (X2), covering the years 2004-2005, was used to assess the final performance of the trained model.

This period was chosen because of the large number of observations covering the entire NAACOM, with data available for all seasons and months. Since these two years of data were not included in the model training and independent test to ensure its independence, it provides an unbiased evaluation of the model's performance on regions or periods without observations. Finally, the trained model is applied to all satellite and reanalysis data to generate the final gap-free reconstructed $fCO_2$ data.



As most products reported seawater $CO_2$ concentration as $pCO_2$, our final product reports $fCO_2$ and $pCO_2$ both, with the $fCO_2$
values being converted to $pCO_2$ using the following equation (Takahashi et al., 2019):

$$pCO_2 = fCO_2 \times (1.00436 - 4.669 \times 10^{-5} \times SST) \tag{2}$$

**2.3 Regression model input variables from satellite and reanalysis**

The input variables for training the regression model include longitude (lon), latitude (lat), month, sea surface temperature
(SST), sea surface salinity (SSS), sea surface height (SSH), and atmospheric $pCO_2$ ($pCO_{2air}$). Longitude, latitude, and month
serve as spatiotemporal predictors, enabling the algorithm to identify and capture regional and seasonal variability in $fCO_2$
within the study area (Su et al., 2020; Yang et al., 2024). SST, SSS, and SSH are critical variables that characterize the physical
and biogeochemical ocean settings, which play a crucial role in determining the spatial and temporal variability of $fCO_2$. The
$pCO_{2air}$ represents the atmospheric forcing on the sea surface $fCO_2$, as the difference between atmospheric and sea surface
$fCO_2$ (or $pCO_2$) primarily determines the direction and magnitude of the air-sea $CO_2$ exchange.

SST data were obtained from the National Oceanic and Atmospheric Administration (NOAA) Optimum Interpolation Sea
Surface Temperature (OISST) v2.1 product (Huang et al., 2021). The OISST dataset is a global gridded SST analysis that
blends observations from various sources, including satellites, ships, and buoys. The dataset employs an optimum interpolation
technique to combine these observations and generate a daily SST field at a spatial resolution of $0.25° \times 0.25°$. For this study,
the daily SST data were averaged to create a monthly product.

SSS data were obtained from the Simple Ocean Data Assimilation (SODA) v3.15.2 product (Carton et al., 2018). SODA is a
comprehensive reanalysis dataset that integrates a global ocean model with observational data to estimate ocean state variables
consistently. The SODA system assimilates observations from multiple sources, including floats, moorings, and ship-based
measurements, thereby constraining the model output and enhancing the accuracy of represented ocean physical properties,
including SSS. The SODA v3.15.2 product offers monthly SSS data with a temporal resolution of one month and a spatial
resolution of $0.5° \times 0.5°$, which were linearly interpolated to a $0.25° \times 0.25°$ grid resolution to maintain consistency with other
input variables and the gridded SOCAT $fCO_2$ data.

SSH data were extracted from the Global Ocean Gridded L4 Sea Surface Heights (European Union-Copernicus Marine
Service, 2021) created by the Copernicus Marine Environment Monitoring Service (CMEMS). This product provides daily
SSH data derived from altimeters, with a spatial resolution of $0.25° \times 0.25°$ since 1993 (ongoing). Daily SSH data were
averaged to monthly means.



$pCO_{2air}$ data converted from the mole fraction of $CO_2$ in the dry air ($xCO_{2air}$) were downloaded from the NOAA Marine Boundary Layer (MBL) reference product (Dlugokencky and Tans, 2022). The MBL reference provides weekly zonal average $xCO_{2air}$ measurements from a global observation network. The $xCO_{2air}$ data was linearly interpolated to the same spatial and temporal resolution as the other input variables ($0.25° \times 0.25°$, monthly). $xCO_{2air}$ was converted to $pCO_{2air}$ with the equation:

$$pCO_{2air} = xCO_{2air} \times (P - p_w)$$     (3)

where $P$ is the atmospheric $CO_2$ pressure at the sea surface, which was downloaded from the fifth generation European Centre for Medium-Range Weather Forecasts (ECMWF) reanalysis (ERA5) (Hersbach et al., 2019), and $p_w$ is the water vapor pressure, which was calculated using the formula of Weiss & Price (1980) using SST from OISST and SSS from SODA.

**2.4 Evaluation of model**

The accuracy of the model outputs was assessed using several statistical metrics, including the coefficient of determination ($R^2$), root mean square error (RMSE), mean absolute error (MAE), and mean bias error (MBE). These metrics were calculated for the training and validation set phases, as well as for the independent validation set:

$$R^2 = 1 - \sum_i^N (y_{obs,i} - y_{est,i})^2 / \sum_i^N (y_{obs,i} - \overline{y_{obs}})^2$$     (4)

$$RMSE = \sqrt{\frac{1}{N} \sum_i^N (y_{obs,i} - y_{est,i})^2}$$     (5)

$$MAE = \frac{1}{N} \sum_i^N |y_{obs,i} - y_{est,i}|$$     (6)

$$MBE = \frac{1}{N} \sum_i^N (y_{obs,i} - y_{est,i})$$     (7)

where $i$ denotes the $i$-th sample, $y_{obs}$ and $\overline{y_{obs}}$ are the observed $pCO_2$ values from SOCAT and their average, $y_{est}$ represents the predicted $pCO_2$ values from the final model, and N is the total number of matched samples.

**2.5 Uncertainty of reconstructed $pCO_2$**

The uncertainty of estimated $pCO_2$ in our product for each grid cell was accumulated from four sources of uncertainties: the direct $pCO_2$ measurement uncertainty from SOCAT ($u_{obs}$), gridding uncertainty ($u_{grid}$), mapping uncertainty ($u_{map}$), and the uncertainty accumulated from the input variables ($u_{inputs}$). The first three sources of uncertainty were calculated according to the approach used by earlier reconstructed $pCO_2$-products (Landschützer et al., 2014; Roobaert et al., 2024a; Sharp et al.,



2022). $u_{obs}$ is inherited from the SOCAT observations. The SOCAT database uses discrete samples with quality flags A and B (accuracy < 2 µatm), and C and D (accuracy < 5 µatm) to create the gridded file. Adopting a conservative approach, we used the maximum $u_{obs}$ of 5 µatm. $u_{grid}$ was calculated as the standard deviation of the samples used to calculate the gridded $fCO_2$ in each grid cell. $u_{map}$ is introduced by reconstructing the $pCO_2$ using the RFR-LR model. It was evaluated as the RMSE between the reconstructed $pCO_2$ and the observed $pCO_2$ values following Roobaert et al. (2024a) and Sharp et al. (2022). Given that the derivation of $u_{obs}$, $u_{grid}$, and $u_{map}$ is contingent upon SOCAT observations, these three uncertainties and the total uncertainty $u_{pCO_2}$ is reported on a sub-regional basis.

In addition to these three sources of uncertainty, this study incorporated cumulative uncertainties from input variables ($u_{inputs}$), including SST, SSS, SSH, and $pCO_{2air}$. These satellite-derived or reanalysis-based variables inherently possess uncertainties that propagate nonlinearly through the regression model, ultimately affecting the estimated $pCO_2$ values (Wang et al., 2021, 2023). We employed a Monte Carlo simulation to calculate $u_{inputs}$. For each input variable (SST, SSS, SSH, $pCO_{2air}$), we added white noise following a normal distribution $N(0, u_{x_i})$, where $u_{x_i}$ is the uncertainty of the respective input variable $x_i$. We then recalculated $pCO_2$ using these noise-added inputs and determined the resulting changes in $pCO_2$. This process was repeated 100 times for each input variable, and the resulting uncertainty in $pCO_2$ from each variable was calculated as the standard deviation of the differences between the original reconstructed $pCO_2$ and the $pCO_2$ values after adding noise in each grid cell. The final $u_{inputs}$ was computed as the square root of the quadratic sum of these individual uncertainties from the four input variables. Detailed procedures for determining $u_{inputs}$ are described in **Appendix A.**

Assuming these sources are independent, the uncertainty of the estimated gridded $pCO_2$ in our product, $u_{pCO_2}$, was calculated using the error propagation (Hughes and Hase, 2010; Taylor, 1997):

$$u_{pCO_2} = \sqrt{u_{obs}^2 + u_{grid}^2 + u_{map}^2 + u_{inputs}^2} \tag{8}$$

**2.6 Comparison with global reconstructed $pCO_2$-product**

The ReCAD-NAACOM-$pCO_2$-product was evaluated through comparisons with seven reconstructed $pCO_2$-products that are developed for the global ocean and used in the Global Carbon Budget 2023 edition (Friedlingstein et al., 2023) and one reconstructed $pCO_2$-product that is specifically developed for the global coastal ocean (Roobaert et al., 2024a). Those data products reconstructed $pCO_{2sea}$ data using different machine-learning algorithms. Detailed information on those products is summarized in Table 1.





**Table 1. References for global $pCO_2$-products used to compare with ReCAD-NAACOM-$pCO_2$ in this study.** The abbreviations in the Methods column are: RFRE for random forest-based Regression Ensemble, SOM-FFN for Self-Organizing Map-Feed Forward Network, MLR for Multiple Linear regression, FFNN for Feed-Forward Neural Network, XGB for eXtreme Gradient Boosting algorithm, and GRaCER for Geospatial Random Cluster Ensemble Regression.

| Data | | Methods | Period | Resolution | Source |
|---|---|---|---|---|---|
| Open ocean product | MPI_SOM-FFN_v2022 | SOM-FNN | 1982-2021 | 1°×1°, monthly | Landschützer et al. (2017) |
| | Jena-MLS | MLR | 1951-2021 | 2°×2°, monthly | Rödenbeck et al. (2022) |
| | CMEMS-LSCE-FFNNv2 | Ensemble of nonlinear models | 1985-2021 | 1°×1°, monthly | Chau et al. (2022) |
| | LDEO-HPD | XGB | 1985-2018 | 1°×1°, monthly | Gloege et al. (2022) |
| | NIES-NN | FFNN | 1980-2020 | 1°×1°, monthly | Zeng et al. (2014) |
| | JMA-MLR | MLR | 1998-2022 | 1°×1°, monthly | Iida et al. (2021) |
| | OS-ETHZ-GRaCER | GRaCER | 1982-2020 | 1°×1°, monthly | Gregor & Gruber (2021) |
| Coastal ocean product | ULB–SOM–FFN–coastalv2 | SOM-FNN | 1982-2020 | 0.25°×0.25°, monthly | Roobaert et al. (2024a) |

# 3 Results and discussion

## 3.1 Evaluating the regression model performance

The ReCAD-NAACOM-$pCO_2$-product demonstrated robust performance and high accuracy in capturing $pCO_2$ variability across the NAACOM (**Fig. 4**). During the model training phase, using 10-fold cross validation, the product achieved an $R^2$ of 0.83, an RMSE of 18.2 µatm, an MAE of 11.58 µatm, and an MBE of 0.17 µatm (**Fig. 4a**). The model demonstrated comparable performance metrics during the validation phase (**Fig. 4b**). To further evaluate the model's generalizability and robustness, we

also conducted an independent test using data from 2004 to 2005, in which all data samples were not included in the model training and validation sets. During this independent test phase, the $pCO_2$-product maintained high accuracy, with $R^2 = 0.65$, RMSE = 26.9 µatm, MAE = 18.81 µatm, and MBE = 0.48 µatm **(Fig. 4c)**. Additionally, most independent validation samples were distributed around the 1:1 corresponding line**,** proving the model's ability to predict $pCO_2$ across unsampled spatial and temporal domains without overfitting. The model consistently demonstrated strong performance during the training, validation,

and independent test phases across all sub-regions **(Table 2)**. Overall, compared with all available samples in SOCAT, it achieved an $R^2$ of 0.83, an RMSE of 18.64 µatm, an MAE of 11.88 µatm, and an MBE of 0.11 µatm for the entire NAACOM, highlighting the ReCAD-NAACOM-$pCO_2$-product's generalizability, and robustness in effectively capturing the variability in $pCO_2$ and providing reliable predictions of $pCO_2$ across the studied regions.



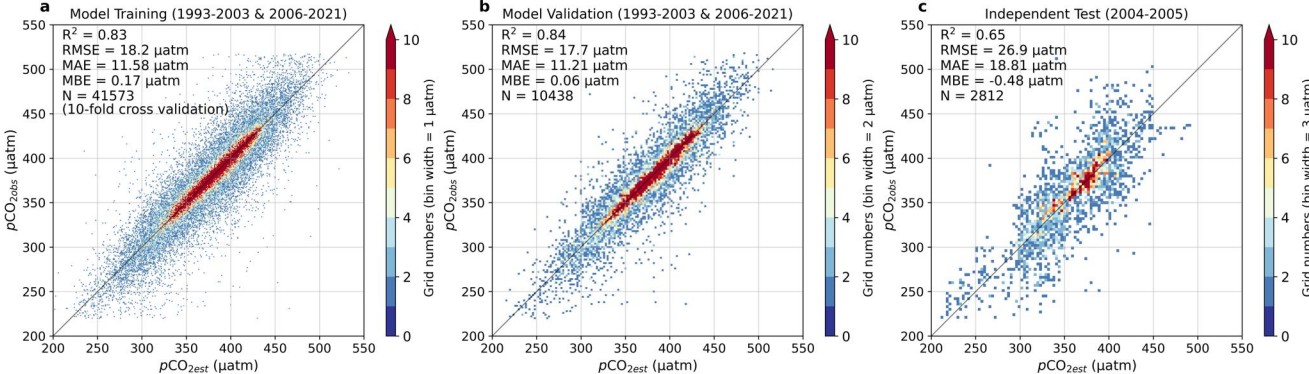

**Figure 4. Evaluation of regression model for reconstructing ReCAD-NAACOM-*p*CO₂ product.** Density scatter plots compare the product estimated $pCO_2$ ($pCO_{2est}$) with the in situ SOCAT observations ($pCO_{2obs}$) during the **(a)** model training phase (80% samples during the period of 1993-2003 and 2006-2021), **(b)** validation phase (20% samples during the period of 1993-2003 and 2006-2021), and **(c)** independent test phase (samples during the period of 2004-2005). The model-estimated values shown in panel (a) were obtained through 10-fold cross-validation. Statistical metrics include the coefficient of determination ($R^2$), root mean square error (RMSE), mean absolute error (MAE), mean bias error (MBE), and the number of samples (N). The color bar represents the number of data points within each bin.



**Table 2. Performance of the regression model during the model training, validation, and independent test phases across different sub-regions.** The model-estimated values during the model training phase were obtained through 10-fold cross-validation. The metrics include the coefficient of determination ($R^2$), root mean square error (RMSE), mean absolute error (MAE), and mean bias error (MBE). Sub-regions are the Gulf of Mexico (GoMx), South Atlantic Bight (SAB), Mid-Atlantic Bight (MAB), Gulf of Maine (GoMe), Scotian Shelf (SS), and Gulf of St. Lawrence and Grand Banks (GStL&GB).

| Region | Type | $R^2$ | RMSE (µatm) | MAE (µatm) | MBE (µatm) |
|---|---|---|---|---|---|
| **GStL&GB** | Training set | 0.90 | 14.45 | 9.55 | 0.91 |
| | Validation set | 0.91 | 13.73 | 9.04 | 0.87 |
| | Independent test set | 0.76 | 25.41 | 17.06 | -7.00 |
| | **All** | **0.89** | **15.55** | **10.08** | **0.25** |
| **SS** | Training set | 0.87 | 16.12 | 11.20 | -0.96 |
| | Validation set | 0.86 | 14.96 | 10.03 | -1.72 |
| | Independent test set | 0.52 | 30.20 | 23.24 | -2.96 |
| | **All** | **0.83** | **17.85** | **12.19** | **-1.30** |
| **GoMe** | Training set | 0.80 | 21.14 | 15.30 | 0.36 |
| | Validation set | 0.80 | 20.54 | 14.99 | 0.01 |
| | Independent test set | 0.49 | 31.98 | 23.99 | 3.99 |
| | **All** | **0.78** | **21.85** | **15.78** | **0.52** |
| **MAB** | Training set | 0.84 | 19.75 | 13.82 | -0.26 |
| | Validation set | 0.86 | 18.93 | 13.35 | -0.10 |
| | Independent test set | 0.59 | 36.80 | 27.86 | 8.24 |
| | **All** | **0.84** | **20.15** | **14.05** | **-0.03** |
| **SAB** | Training set | 0.87 | 12.28 | 7.50 | 0.54 |
| | Validation set | 0.89 | 11.36 | 6.93 | 0.03 |
| | Independent test set | 0.74 | 23.06 | 16.79 | -0.57 |
| | **All** | **0.85** | **13.56** | **8.30** | **0.34** |
| **GoMx** | Training set | 0.77 | 18.61 | 10.11 | 0.04 |
| | Validation set | 0.76 | 18.33 | 9.83 | 0.15 |
| | Independent test set | 0.49 | 14.28 | 7.60 | -4.31 |
| | **All** | **0.77** | **18.46** | **10.00** | **-0.05** |
| **NAACOM** | Training set | 0.83 | 18.19 | 11.58 | 0.17 |
| | Validation set | 0.84 | 17.67 | 11.21 | 0.06 |
| | Independent test set | 0.65 | 26.93 | 18.81 | -0.48 |
| | **All** | **0.83** | **18.64** | **11.88** | **0.11** |

Earth System
Science
Data

## 3.2 Spatial distribution of product bias

The ReCAD-NAACOM-$p$CO$_2$ product exhibited a negligible area-mean bias of +0.17 µatm with a standard deviation of 9.48

µatm when compared to all SOCAT observation grid cells across the entire NAACOM (**Fig. 5**). This small average difference

suggests no consistent over- or under-estimation by the regression model, indicating the product's reliability in estimating the

monthly and annual mean climatology of $p$CO$_2$ across the entire NAACOM region.

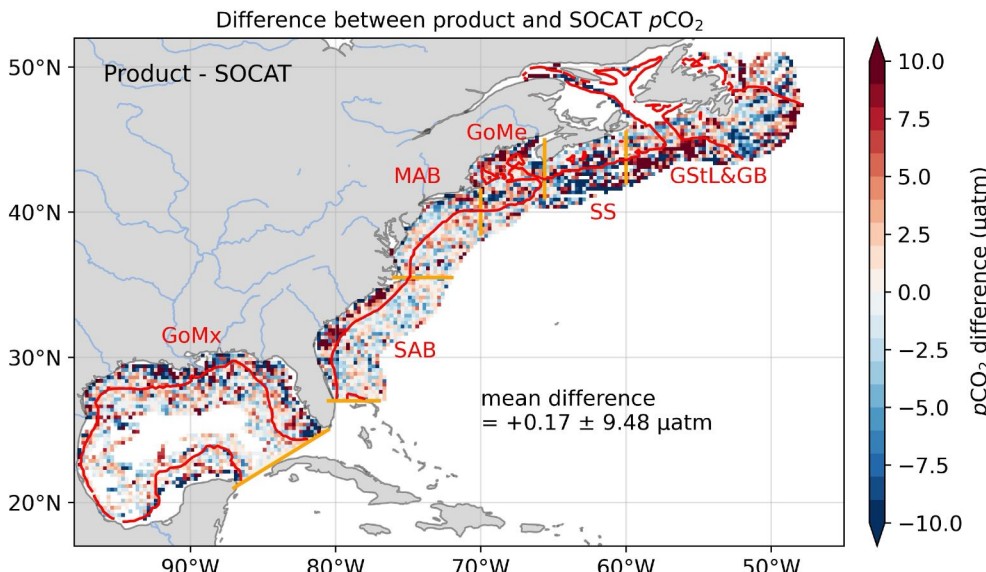

**Figure 5. Spatial distribution of mean bias error (MBE) between ReCAD-NAACOM-$p$CO$_2$ product and SOCAT**
**observations across the NAACOM.** The MBE is calculated for each grid cell as the average difference between product
estimates and SOCAT observations. Positive values (red) indicate product overestimation, while negative values (blue)
indicate underestimation relative to SOCAT. The overall mean difference is +0.17 ± 9.48 µatm. The NAACOM is divided into
six sub-regions by the orange straight lines, including the Gulf of Mexico (GoMx), South Atlantic Bight (SAB), Mid-Atlantic
Bight (MAB), Gulf of Maine (GoMe), Scotian Shelf (SS), and Gulf of St. Lawrence and Grand Banks (GStL&GB). The
contour line in red is the 200 m isobath, which is roughly the location of the shelf break and a typical definition of continental
shelf boundary.

While the area-average difference is small, the differences are distributed heterogeneously in space. Larger differences

(absolute difference > 10 µatm) tend to occur in nearshore regions, particularly along the coastlines of the GoMx and SAB, as

well as in northern areas such as the GoMe, SS, and GStL&GB (**Fig. 5**). These regional variations can be attributed to complex

coastal processes such as terrestrial inputs, sparse observations in the northern areas (Lavoie et al., 2021; Rutherford et al.,

2021; Salisbury and Jönsson, 2018), and less accurate satellite observations in the nearshore regions (Song et al., 2023). Conversely, smaller differences (absolute difference < 2.5 µatm) are observed in the central parts of the GoMx, offshore regions of the SAB and MAB, and some nearshore regions of the SS and GB, which is likely due to more stable oceanic conditions in those regions. Despite these regional differences, the product's small overall difference underscores its effectiveness in capturing the broader $pCO_2$ patterns across the NAACOM.

### 3.3 Evaluating the product's capacity to capture $pCO_2$ seasonality

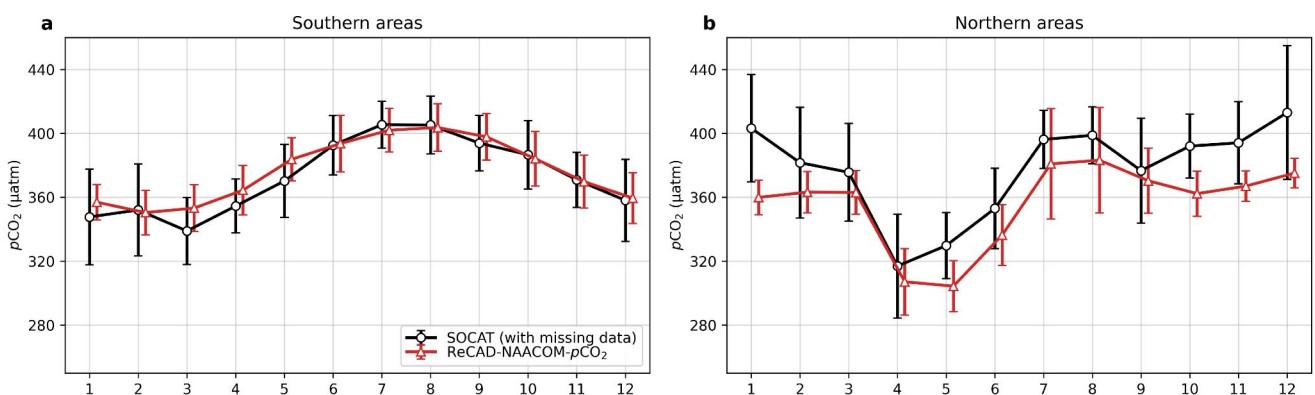

**Figure 6. Monthly mean climatology of $pCO_2$ in the southern and northern areas of the NAACOM from 1993 to 2021.** Sub-regions are **(a)** southern areas, the red box in Fig. 2, and **(b)** northern areas, the blue box in Fig. 2. The x-axis represents months (1-12 with 1 equal January), and the y-axis shows $pCO_2$ in µatm. Two data representations are shown: (1) SOCAT observations (black curves), which may be influenced by missing data; and (2) the complete product output (red curves). Error bars denote one standard deviation of the monthly mean climatology of $pCO_2$.

**Figure 6** showcases the applicability of the product in capturing the $pCO_2$ seasonal cycles across the southern and northern areas of NAACOM. The two data representations show strong similarity, demonstrating that ReCAD-NAACOM-$pCO_2$-product effectively captures the seasonal cycles of $pCO_2$ in the diverse coastal environments of the NAACOM region. Our product reveals distinct seasonal cycles between the southern (the red box in Fig. 2) and northern areas (the blue box in Fig. 2). The seasonal cycle of $pCO_2$ in the southern areas exhibits maximum $pCO_2$ in summer (July and August, around 400 µatm) and minimum in late winter to early spring (January to March, around 340 µatm) **(Fig. 6a)**. In contrast, the northern areas display minimum $pCO_2$ in late spring (April and May, less than 320 µatm) and maximum in summer (July and August, around 380 µatm) **(Fig. 6b)**. The seasonal $pCO_2$ amplitude, defined as the difference between maximum and minimum monthly mean $pCO_2$ values within a year (Takahashi et al. 2002), is smaller in the southern areas than the northern areas, with mean amplitude of around 60 µatm in the south **(Fig. 6a)** but circa a twofold increase to 110 µatm in the north **(Fig. 6b)**. A notable feature





revealed by the product is the elevated $pCO_2$ values during fall and winter in the northern areas. While $pCO_2$ decreases after the summer peak in southern regions, northern areas maintain relatively high $pCO_2$ levels throughout fall and winter **(Fig. 6b)**.

The product predicts smaller monthly standard deviations in southern regions (less than 40 µatm, error bars in **Fig. 6a**), suggesting higher model accuracy and less interannual variability in these areas. Conversely, larger monthly standard deviations are observed in the northern areas, suggesting potential less accuracy and remarkable interannual variability. However, the larger interannual variability in these areas may be an artifact due to the limited observational data available for regression model training, resulting in greater uncertainty in the predictions. Despite differences in the mean monthly

climatology, the similar seasonal $pCO_2$ cycles calculated from SOCAT and reconstructed product demonstrate the ReCAD-NAACOM-$pCO_2$-product's capability to represent seasonal $pCO_2$ variability across diverse coastal environments. Nevertheless, there exist larger differences between the observations and reconstructed $pCO_2$ in some months and regions **(Fig. 6b)**, highlighting the importance of the gap-free product in an unbiased understanding of regional carbon cycles (Ren et al., 2024). Detailed sea surface $pCO_2$ seasonal cycles and their controlling mechanism across different sub-regions of the NAACOM will

be presented in our subsequent work.

### 3.4 Evaluating the product's ability to capture regional variation by comparing it to global products

The ReCAD-NAACOM-$pCO_2$ product demonstrates the capability to resolve fine-scale regional spatial distributions of $pCO_2$. **Figure 7** illustrates the spatial distribution of annual mean climatology of $pCO_2$ across the NAACOM as observed by SOCAT and predicted by different global open and coastal $pCO_2$-products. Despite being affected by missing data, SOCAT

observations **(Fig. 7a)** reveal significant regional variations in $pCO_2$, such as the low $pCO_2$ levels (<340 µatm) in the Louisiana Shelf (LAS) estuary plume region (box 1 in Fig. 7) and relatively higher values (> 400 µatm) in the West Florida Shelf (WFS, box 2 in Fig. 7), which have been systematically reported in earlier studies (Kealoha et al., 2020; Robbins et al., 2018; Wu et al., 2024b). The ReCAD-NAACOM-$pCO_2$ product demonstrates superior alignment with SOCAT observations in capturing these regional features **(Fig. 7b)**, accurately representing the low $pCO_2$ values in the LAS Mississippi River plume (box 1) and

the elevated $pCO_2$ levels in the WFS (box 2), underscoring the product's capacity to resolve regional spatial variations in coastal $pCO_2$ dynamics.

In contrast, the global reconstructions of $pCO_2$, represented by the ensemble of the seven open ocean $pCO_2$-products **(Fig. 7c)**, face challenges in resolving these regional $pCO_2$ variations, as previously discussed by Wu et al (2024b). The coastal $pCO_2$-product of Roobaert et al. (2024a, ULB_SOMFFN_coastal_v2) also captures some small-scale structures, like low $pCO_2$ in

the LAS **(Fig. 7d),** but the ReCAD-NAACOM-$pCO_2$ product exhibits closer values to the observations. In the northern area (box 3), the ReCAD-NAACOM-$pCO_2$ product predicts higher $pCO_2$ levels that are closer to observations in the nearshore



region (**Fig. 7b**). This is not surprising, as the ULB_SOMFNN_coastal_v2 is a global product known for its high accuracy on the global average. These comparisons highlight the necessity of developing regional reconstructed products in capturing the complex spatial heterogeneity of coastal $pCO_2$ distributions. The ReCAD-NAACOM-$pCO_2$ product's ability to capture

regional features suggests its potential utility for studies focusing on coastal carbon dynamics and their response to local and regional forcing factors in future research in the NAACOM.

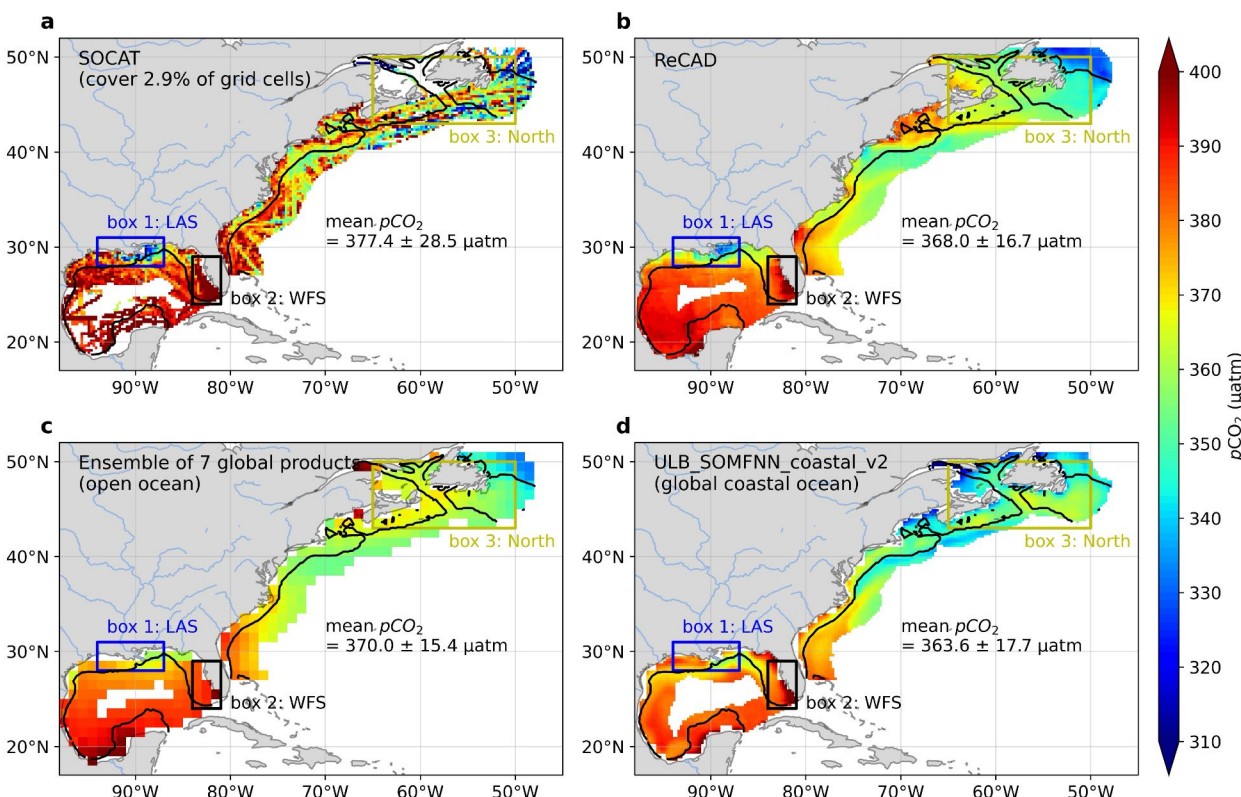

**Figure 7. Spatial distribution of annual mean $pCO_2$ climatology in the NAACOM from different sources. (a)** SOCAT observations, **(b)** ReCAD-NAACOM-$pCO_2$ product, **(c)** Ensemble mean of 7 global open ocean $pCO_2$-products listed in Table

1, and **(d)** Coastal $pCO_2$-product ULB_SOMFNN_coastal_v2 (Roobaert et al., 2024a). The black contour delineates the coastal ocean margin. Three boxes represent sub-regions in the NAACOM: box 1 for the Louisiana Shelf (LAS), box 2 for the West Florida Shelf (WFS), and box 3 for the Northern region. Mean $pCO_2$ values ± standard deviation of all grid cells are provided for each dataset. Color scale represents $pCO_2$ in µatm.





### 3.5 Evaluating the product's capacity to detect decadal linear trends of $pCO_2$

Using $pCO_2$-products to accurately reconstruct $pCO_2$ linear trends in coastal regions presents significant challenges due to the high spatial heterogeneity of coastal $pCO_2$ dynamics. This heterogeneity often leads to sea surface $pCO_2$ changes that deviate from atmospheric trends (Laruelle et al., 2018). Even when utilizing similar observational datasets, derived products may not consistently reflect the underlying trends. For instance, Wu et al. (2024b) examined the capability of various products to reflect $pCO_2$ changes in the GoMx, a region where $pCO_2$ trends exhibit significant spatial variability. Despite this heterogeneity,

seven global open ocean products (listed in table **Table 1**) indicate trends similar to atmospheric $pCO_2$ across the entire GoMx without regional differences. In contrast, the GoMx-specific regional product developed by Chen and Hu (2019) demonstrates no significant overall trend. The discrepancy in trend detection primarily stems from the design of the regression model and the selection of input variables. These factors are critical in capturing the complex spatiotemporal variability of coastal $pCO_2$ and its long-term evolution.

To assess the product's capability in resolving decadal $pCO_2$ trends, we conducted an analysis of $pCO_2$ evolution using three distinct regions within the NAACOM (three boxes in Fig. 7) as representative examples (**Fig. 8**). Decadal trends of deseasonalized time series were calculated following the established protocol described by Sutton et al. (2022). The LAS (box 1 in **Fig. 7**) has been identified as an increasing $CO_2$ sink, characterized by a negative $pCO_2$ rate increase from 2002-2021 (Wu et al., 2024). Our product results for the extended period of 1993-2021 indicate that $pCO_2$ increased at a rate of $+0.32 \pm 0.11$

$\mu atm\ yr^{-1}$ (**Fig. 8a**). This rate is significantly lower than the observed atmospheric $pCO_2$ increase in this region, which is approximately $+1.8\ \mu atm\ yr^{-1}$. These findings corroborate our previous conclusion that the LAS is an increasing $CO_2$ sink, demonstrating our product's capability to reveal long-term $pCO_2$ trends in this dynamic river plume region, extending the analysis period by nearly a decade compared to previous studies. In contrast, the WFS (box 2 in **Fig. 7**) exhibits accelerated $pCO_2$ increase faster than atmospheric $pCO_2$ of around $+2.0\ \mu atm\ yr^{-1}$ (**Fig. 8b**), aligning with observations reported by Robbins

et al. (2018), which found a transition from a $CO_2$ sink to a source in this region during the 1990s.

  Both ReCAD-NAACOM-$pCO_2$ and SOCAT consistently report a $pCO_2$ trend around $+2.30\ \mu atm\ yr^{-1}$ in the northern area (box 3 in **Fig. 7**) over 1993-2021 (**Fig. 8c**), which is faster than the atmospheric $pCO_2$ increase (around $+2.0\ \mu atm\ yr^{-1}$), suggesting that these areas have been becoming a decreasing $CO_2$ sink. However, limited observational data in this area necessitates cautious interpretation and warrants further validation in future research. Overall, the spatiotemporal heterogeneity in surface

ocean $pCO_2$ trends across the NAACOM underscores the importance of long-term monitoring to elucidate the drivers of these trends, particularly in regions influenced by major current systems and in areas with limited observational data.

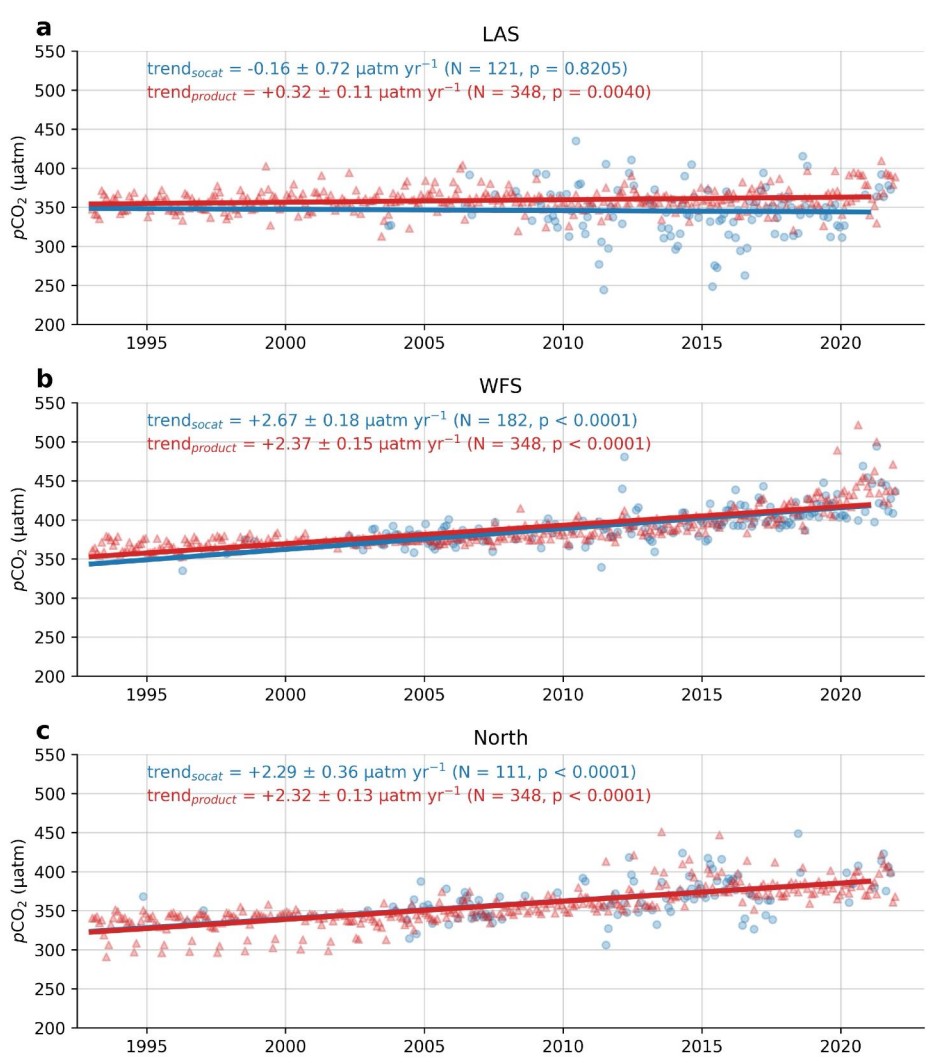

**Figure 8. Decadal linear trends of sea surface $pCO_2$ in three regions of the NAACOM from 1993-2021.** Blue and red dots are monthly average $pCO_2$ values (deseasonalized) calculated from SOCAT observations and reconstructed ReCAD-NAACOM-$pCO_2$, respectively. Thick lines are linear fitted regression lines. Three regions are the boxes in Fig 7: **(a)** Louisiana Shelf (LAS), northern Gulf of Mexico shelf river plume region; **(b)** West Florida Shelf (WFS); and **(c)** Northern areas. Linear trends are calculated following the established protocol by Sutton et al. (2022). Numbers in parentheses are the number of months with data and the p values.






### 3.6 Evaluating the product's uncertainty

**Table 3. Uncertainty estimates for the ReCAD-NAACOM-$p$CO$_2$ product across different sub-regions of the NAACOM.**
$u_{obs}$, $u_{grid}$, $u_{map}$, and $u_{inputs}$ represent the measurement uncertainty, gridding uncertainty, mapping uncertainty, and uncertainty accumulated from input variables, respectively (see method Section 2.5 for further details). $u_{pCO2}$ is the total combined uncertainty. All values are in μatm. Sub-regions are the Gulf of Mexico (GoMx), South Atlantic Bight (SAB), Mid-Atlantic Bight (MAB), Gulf of Maine (GoMe), Scotian Shelf (SS), and Gulf of St. Lawrence and Grand Banks (GStL&GB).

| Region | $u_{obs}$ | $u_{grid}$ | $u_{map}$ | $u_{inputs}$ | $u_{pCO2}$ |
|---|---|---|---|---|---|
| GStL&GB | 5.00 | 15.44 | 15.55 | 5.57 | 23.16 |
| SS | 5.00 | 15.37 | 17.85 | 6.18 | 24.86 |
| GoMe | 5.00 | 16.05 | 21.85 | 7.51 | 28.57 |
| MAB | 5.00 | 16.14 | 20.15 | 5.97 | 26.97 |
| SAB | 5.00 | 8.29 | 13.56 | 5.99 | 17.70 |
| GoMx | 5.00 | 10.38 | 18.46 | 5.55 | 22.45 |
| NAACOM | 5.00 | 12.69 | 18.64 | 5.86 | 23.83 |

Uncertainty of the reconstructed $p$CO$_2$-product was estimated by accumulating uncertainties from mapping ($u_{map}$), gridding ($u_{grid}$), measurement ($u_{obs}$), and input variables ($u_{inputs}$, see Section 2.5 of the method for further details on the calculation). To maintain a conservative estimate, we adopted the larger value of 5 μatm as $u_{obs}$ for all data points. The gridded $f$CO$_2$ values from SOCAT are reported as the averages of all samples collected within each grid cell. Accordingly, $u_{grid}$ was quantified as the standard deviation of samples within each grid cell, calculated across six sub-regions. $u_{map}$ was calculated using the RMSE values reported in Table 2 following previous literature (Roobaert et al., 2024a; Sharp et al., 2022). $u_{inputs}$ was calculated using a Monte Carlo simulation (**Appendix A**). These four sources of uncertainties were evaluated across different sub-regions of the NAACOM, as shown in **Table 3**. $u_{map}$ contributes the largest portion to the total uncertainties across all sub-subregions with the maximum value up to 21.85 μatm in the SS. Overall, the ReCAD-NAACOM-$p$CO$_2$ product demonstrates uncertainties ranging between 17 to 29 μatm across six sub-regions, and an average uncertainty of 23.83 μatm for the entire NAACOM.

This uncertainty range is deemed reasonable, considering our conservative estimation approach. For comparison, estimated uncertainties in the North American Pacific Coastal Ocean Margin were 43.4 μatm (Sharp et al., 2022). It is important to note that our uncertainty calculation assumed independence among all sources, which is a simplification. Recent research by e.g., Ford et al. (2024) has highlighted that these uncertainties are often correlated. Future studies should consider these inter-variable correlations to refine uncertainty estimates.

### 3.7 Challenges and Limitations

Even though ReCAD-NAACOM-$p$CO$_2$ resolves regional $p$CO$_2$ variability with high accuracy in the NAACOM, this product still has room for improvement in the future. Potential areas for improvement include the 0.25° spatial resolution, which is inadequate to resolve sub-mesoscale variability at the scale of 0.1 - 10 km (McWilliams, 1985). Furthermore, the performance

of the model during the independent validation phase reduced in the GoMe ($R^2 = 0.49$) and GoMx ($R^2 = 0.49$) **(Table 2)**, which may be due to the complex biological and physical condition in the estuary plume regions in these two gulfs. In this study, we opted not to include chlorophyll-a (Chl-a) concentrations and wind speeds as input variables for model training and prediction. This decision was primarily due to the limited temporal coverage of satellite-derived Chl-a data, which only extends back to 1997 with the launch of the Sea-viewing Wide Field-of-view Sensor (SeaWiFS) satellite (O'Reilly et al., 1998). The inclusion of Chl-a would have restricted our model's temporal range, potentially limiting its ability to capture long-term trends and variability in $p$CO$_2$. Future versions of our model will aim to address this limitation. One potential approach is to develop a two-phase model: one for the period before 1997 without Chl-a data, and another for the post-1997 period incorporating Chl-a information. Alternatively, we may explore methods to reconstruct historical Chl-a data or use proxy variables that correlate with biological productivity and are available for the entire study period.

In our previous work, we demonstrated that incorporating wind speeds and sea surface roughness data derived from Synthetic Aperture Radar (SAR) could enhance model performance in predicting $p$CO$_2$ at submesoscale resolutions (Wang et al., 2024). In this work, we evaluated the inclusion of wind speed as an input variable in our model. However, at the 0.25° resolution employed here, the addition of wind speed data did not significantly improve model performance (only increase the $R^2$ by 0.1). Moreover, using the same Monte Carlo simulation approach applied to other variables, incorporating wind speeds would introduce an additional 6 µatm uncertainty to $p$CO$_2$ estimates, doubling the input-related uncertainties. Consequently, we excluded wind speeds from our regression model to reduce input-related uncertainties. Despite this omission, our product demonstrates robust capability in resolving regional variations, seasonal cycles, and decadal trends in $p$CO$_2$, making it valuable for future studies.

## 4 Data availability

The reconstructed $f$CO$_2$, $p$CO$_2$, and the uncertainty in ReCAD are available as a NetCDF file at https://doi.org/10.5281/zenodo.11500974 (Wu et al., 2024a) and will be updated regularly.

## 5 Code availability

Python and MATLAB code used to process data and create figures included in this paper is provided at https://github.com/zelunwu/ReCAD_product_v1



## 6 Conclusions

The ReCAD-NAACOM-$p$CO$_2$ product developed in this study represents a significant advancement in our ability to detect the spatial variations, seasonal cycle, and decadal changes of surface ocean $p$CO$_2$ dynamics in the NAACOM. By leveraging a two-step approach combining random forest and linear regression, and a set of environmental predictors, we have created a high-resolution, long-term dataset (1993-2021 period) that captures the complex spatial and temporal variability of $p$CO$_2$ across the region. On average, compared with all available samples from the SOCAT observations in our study region, the

product has an R$^2$ of 0.83, an RMSE of 18.64 µatm, an MAE of 11.88 µatm, and an MBE of 0.11 µatm for the entire NAACOM, with an average uncertainty of 23.83 µatm. Key findings from this study include:

1. The product demonstrates high accuracy and reliability, as evidenced by strong performance metrics during training, validation, and independent test phases across six sub-regions.
2. Distinct seasonal cycles are observed between southern and northern sub-regions, with the product capturing nuanced
features such as elevated $p$CO$_2$ levels during fall and winter in northern areas.
3. Comparison with global products highlights the superior ability of the ReCAD-NAACOM-$p$CO$_2$ product to resolve fine-scale coastal features and variability.
4. The $p$CO$_2$-product successfully reconstructed decadal linear trends consistent with previous studies, while also revealing a rapid increase in $p$CO$_2$ in the northern regions of the NAACOM.

While areas for future improvement exist, such as increasing spatial resolution and enhancing accuracy in estuary plume-influenced regions, the ReCAD-NAACOM-$p$CO$_2$ product provides a robust foundation for studying coastal carbon dynamics. This dataset will be valuable for investigating air-sea CO$_2$ fluxes, assessing ocean acidification impacts, and understanding the role of coastal systems in the NAACOM.

Future research should validate the reconstructed trends, particularly in areas with limited observational data, and explore the
mechanisms driving the spatiotemporal variability in $p$CO$_2$ across the NAACOM region. Additionally, the methodologies developed here can contribute to a more comprehensive understanding of coastal ocean carbon dynamics in the face of climate change and have the potential to be applied globally.



**Appendix A: Monte Carlo simulation in calculating $u_{inputs}$**

A crucial step in calculating $u_{inputs}$ is determining the uncertainties of the input variables. In our reconstructed model, there
were four variables that need to be evaluated: SST, SSS, SSH, and $pCO_{2air}$. Our general principle was to adopt conservative
estimates, using the largest reported uncertainty for each product when available.

SST errors are provided within the OISST product at the grid level. On the global average, OISST reports a mean bias and
RMSE of -0.04 and 0.24 °C when compared with the observations on the global average (Huang et al., 2021). For our study
region, we calculated the mean SST error across all grid cells, yielding a value of 0.23°C.

The SODA database assimilates observational data but does not directly provide SSS error estimates. Given this limitation in
uncertainty reporting, we derived an estimate based on the RMSE between model SSS and observations near our study region,
as reported by Carton et al. (2018). Their analysis (their Fig. 8) indicates an RMSE exceeding 0.3 psu in the vicinity of our
area of interest. To maintain a conservative approach in our uncertainty quantification, we doubled the uncertainty and adopted
a value of 0.6 psu as the SSS uncertainty for our calculations.

SSH errors are directly provided in the dataset, which has a mean uncertainty of 1.8 cm in our study region.

$pCO_{2air}$, calculated from $xCO_{2air}$ (MBL References), which has a global mean uncertainty of 0.22 ppm.

To propagate these input uncertainties to the final $pCO_2$ estimate, a Monte Carlo simulation approach was implemented:

1.  For each input variable $x_i$, random perturbations $\varepsilon_i$ were generated following a normal distribution N(0, $u_i$), where $u_i$
    represents the uncertainty of the respective variable listed above.
2.  Perturbed inputs ($x_i + \varepsilon_i$) were used to calculate $pCO_2$ using the established model.
3.  The difference ($\Delta_i$) between the reconstructed $pCO_2$ before and after adding the perturbation was computed.
4.  Steps 1, 2, and 3 were iterated 100 times for each input variable.
5.  The uncertainty contribution from each variable was quantified as the standard deviation of the 100 $\Delta_i$ values in each
    grid cell.

The total uncertainty attributed to input variables ($u_{inputs}$) was then calculated as the square root of quadratic sum of individual
uncertainties:

$$u_{inputs} = \sqrt{u_{SST}^2 + u_{SSS}^2 + u_{SSH}^2 + u_{pCO_{2air}}^2} \qquad (A1)$$

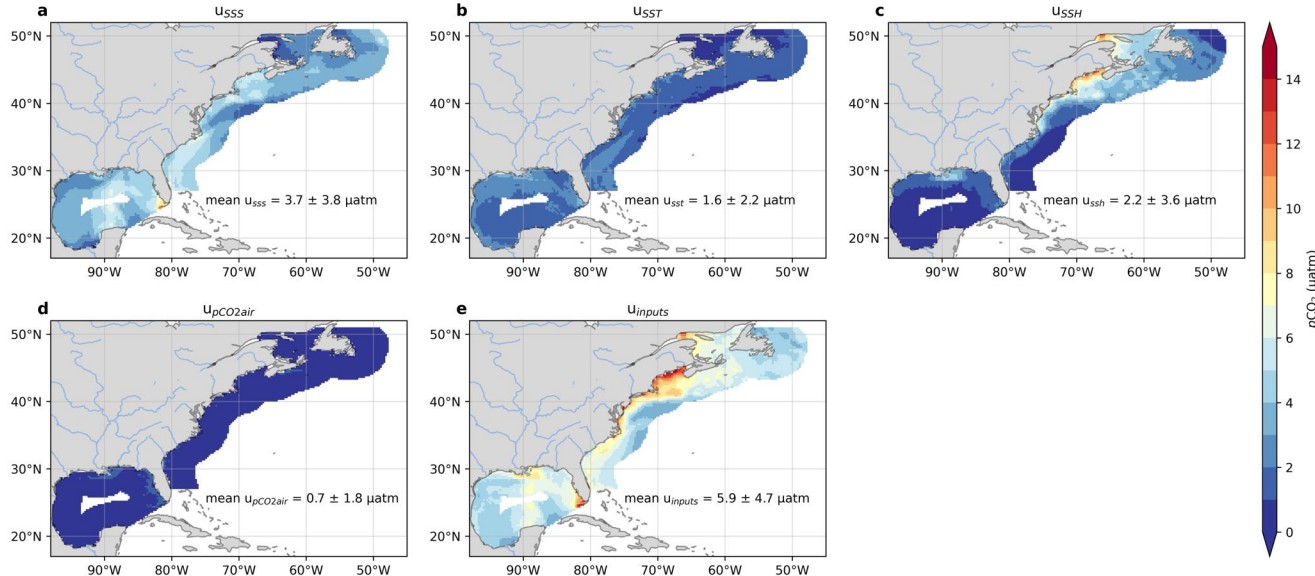

**Figure A1. Uncertainties of $p$CO$_2$ accumulated from different input variables for the model.**

The largest uncertainties propagated from these variables are sourced from SSS and SSH **(Fig. A1a and A1c).** Simulating salinity in coastal regions are still challenging due to complex lang-ocean interaction. For the SSH, the largest uncertainties were observed in the GoMe and GStL. Overall, $u_{inputs}$ is largest in the West Florida Shelf and nearshore waters around the GoMe, with a mean $u_{inputs}$ uncertainty of $5.9 \pm 4.7$ µatm for the entire NAACOM.

## Author contribution

**Zelun Wu**: Conceptualization, data curation, formal analysis, methodology, software, visualization, writing – original draft preparation, writing – review & editing. **Wenfang Lu**: Funding acquisition, methodology, validation, writing – review & editing. **Alizée Roobaert**: Validation, writing – review & editing. **Luping Song**: Validation, writing – review & editing. **Xiao-Hai Yan**: Project administration, supervision. **Wei-Jun Cai**: Conceptualization, project administration, supervision, validation, writing – review & editing.

## Competing interests

The authors declare that they have no conflict of interest.

## Disclaimer

## Special issue statement

## Acknowledgments

The authors thank the NOAA, the Remote Sensing System, and the carbonate community (SOCAT) for sharing their data. This work is part of Zelun Wu's Ph.D. Dissertation under the University of Delaware-Xiamen University Dual Degree Program in Oceanography.

## Financial support

This research has been supported by the Southern Marine Science and Engineering Guangdong Laboratory (Zhuhai) (No. SML2023SP238) to Wenfang Lu, and Industry-University Cooperation and Collaborative Education Projects (202102245034) and PhD Fellowship of the State Key Laboratory of Marine Environmental Science at Xiamen University to Zelun Wu.

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
