# Peer review of "A machine-learning reconstruction of sea surface *p*CO2 in the North American Atlantic Coastal Ocean Margin from 1993 to 2021"

_Earth System Science Data, 2024_

## Referee Comment (RC1)

**Review of Wu et al. (essd-2024-309)**

**General comments**

Wu et al. describe a new data product that reconstructs sea surface $pCO_2$ in the North American Atlantic coastal ocean margin over nearly thirty years. The authors rely on the gridded Surface Ocean $CO_2$ Atlas (SOCAT) dataset as the baseline observations for this data product, and they reconstruct $pCO_2$ using a two-step random forest regression (RFR) + linear regression (LR) approach. They find that their data product (ReCAD-NAACOM-$pCO_2$) effectively captures coastal features and variability along the North Atlantic coastal margin. The authors report a region-wide $R^2$ of 0.83 and RMSE of 18.64 µatm in comparison to observations. Overall, this manuscript describes a useful product that has value for those engaged in studies of ocean acidification and air-sea $CO_2$ flux in the region. There are some areas, however, where more detailed explanations and thorough analyses would make this a stronger contribution.

The strategy of adjusting RFR estimates with an LR is a unique and straightforward way to mitigate possible biases in the RFR estimates. However, this aspect of the methodology could use more explanation, in particular with respect to why this correction might be needed and how it improves the product ReCAD-NAACOM-$pCO_2$ relative to not implementing the LR step. If, as indicated, the LR serves to "mitigate potential systematic biases in RFR-derived $fCO_2$ values [that] arise from spatiotemporal heterogeneities in the SOCAT observational dataset", I envision a figure like Fig. 4c before and after applying the LR would emphasize the added value of this methodological step.

I find the analysis presented in Section 3.3 to be somewhat lacking. While the similarities in large-scale climatological patterns between the raw observations and ReCAD-NAACOM-$pCO_2$ is encouraging, more interesting is where, when, and why the two datasets differ, and how those differences speak to the value added by the gap-filled product. In particular, I see much higher wintertime $pCO_2$ in the observations compared to the product in the northern region in Fig. 6. Is this result due to preferential observational coverage of high-$pCO_2$ areas in that season, as potentially indicated by Fig. 2d? This type of analysis I think is more interesting to readers, and more effective at communicating the utility of the new product.

The comparisons to global products detailed in Section 3.4 would benefit from some quantitative results to be presented alongside the qualitative interpretation of the annual mean climatological figures. The authors assert, for example, that compared to ULB_SOMFFN_coastal_v2 "the ReCAD-NAACOM-$pCO_2$ product exhibits closer values to the observations", but provide no evidence outside visual inspection of Fig. 7. Instead, by comparing (for instance) the average and RMSE of differences between the gridded SOCAT observations and corresponding values from the products within specific regions, the authors could more clearly emphasize the level of improvement provided by ReCAD-NAACOM-$pCO_2$.

Additional line-specific comments are provided below.

**Specific comments**

49: Aren't the $fCO_2$ data included in SOCAT from these cruises exclusively from underway measurements (not discrete)? In which case, perhaps this sentence should read "Underway measurements from these research cruises, combined with underway measurements from volunteer observing ships and buoy observations, …" or something to that effect.

Figure 1: The regional labels might be better displayed in orange rather than red. As they are now, one might understandably associate the red labels with the red 200m isobath, which can be confusing.

75–77: These two sentences say essentially the same thing and could be combined.

105: The word "enhanced" suggests a comparison for the spatial, seasonal, and decadal variability. It should be mentioned here that the capability of the product at resolving these variations is enhanced in reference to some other dataset. Global products? The gridded SOCAT observations?

109: I find the "ground-truth data" terminology to be somewhat misleading. Ground-truth suggests data that is used to evaluate some model or remote-sensing measurement, but here the data is used not only as a ground-truth but also for training the model itself. Perhaps something like "observational data", "model-training data", etc. might be more appropriate.

118: Sampling density also looks to be particularly low in the western Gulf of Mexico.

Figure 3: My understanding is that the "Model" (light orange box with curved sides) is the same that is applied to all satellite and reanalysis data to construct the gridded product. As such, I'd recommend some modification to this flow chart. The arrow from "Model" to "Predictive model" is confusing if those two items are indeed the same.

167–168: More explanation should be given here on exactly how the validation set is used to evaluate the model performance.

171–172: This sentence is somewhat unclear.

178: How is month treated in the model training? If you're only using 1–12 for the months of the year, there will be an unintended effect whereby months that should be treated as similar (e.g., January vs. December) will be treated as extremely different (1 vs. 12). See Sauzède et al. (2015) or Gregor et al. (2018) for information about transforming cyclical predictors using sine and cosine functions.

206: I believe $P$ represents the total atmospheric pressure, not the "$CO_2$ atmospheric pressure".

315–324: I'm not sure this discussion is very valuable because the general features discussed here are evident in the product but also in the observations themselves. It might be more effective to discuss the seasonal cycle features in the product as they relate to the observations; what information is added by the gap-filled product?

416: It should be clarified here that this uncertainty value for the North American Pacific Coastal Ocean Margin is specific to areas within 100km of the coastline and the uncertainty provided for ReCAD-NAACOM- $p$CO$_2$ is for areas within 400km.

**Technical corrections**

148: Should be "arising" or "that arise"

424: Recommend changing wording here: "the performance…reduced" is somewhat awkward

**References**

Gregor, L., Kok, S., & Monteiro, P. M. S. (2018). Interannual drivers of the seasonal cycle of CO2 in the Southern Ocean. Biogeosciences, 15(8), 2361–2378. https://doi.org/10.5194/bg-15-2361-2018

Sauzède, R., Claustre, H., Jamet, C., Uitz, J., Ras, J., Mignot, A., & D'Ortenzio, F. (2015). Retrieving the vertical distribution of chlorophyll a concentration and phytoplankton community composition from in situ fluorescence profiles: A method based on a neural network with potential for global-scale applications. Journal of Geophysical Research: Oceans, 120(1), 451–470. https://doi.org/10.1002/2014JC010355

---

## Author Comment (AC1)

**Response letter to Reviewer#2**

**Overview**

The authors are presenting a new, regional  $pCO_2$ -product specifically designed for the North American Atlantic coastal region that provides monthly  $pCO_2$  at a .25-degree spatial resolution from 1993-2021. The product uses integrated random forest and linear regression methods incorporating observational products in order to generate their monthly reconstructed  $pCO_2$ -product. This allows for analysis of regional, seasonal, and yearly trends in addition to an uncertainty calculation. The authors find through validation that their product provides high accuracy, improving public access for more precise, higher resolution coastal carbon dynamics in the NAACOM region.

There is great need for products like these to be publicly accessible, and this contributes an important resource to the scientific community. The authors do a nice job of introducing the field, the data gaps, and where this product can contribute to those gaps. I do recommend for publication, following a few edits as outlined below.

**General responses:** We sincerely appreciate the reviewer's recognition of the value of our work. We have carefully addressed the reviewer's suggestions for strengthening our contribution through more detailed explanations and thorough analyses. The manuscript has been extensively revised to incorporate feedback from both reviewers. Major improvements include:

- 1. Clarified the usage and distinction between  $fCO_2$  and  $pCO_2$  throughout different sections of the manuscript
- Enhanced the presentation and explanation of Mean Bias Error (MBE) in Section
  3.2 and Fig. 5 to avoid potential confusion
- 3. Substantially revised Section 3.3 to better explain why product-estimated  $pCO_2$  and SOCAT observations show discrepancies in the northern areas, attributing these differences to limited observational coverage
- 4. Restructured Section 3.4 and Fig. 7 to clearly differentiate between previously documented regional variations and newly identified phenomena revealed by our product

Additionally, we have made an important revision regarding model evaluation. In our previous version, we reported the model outputs for the training dataset (80% of X1) using results from 10-fold cross-validation. We have now updated our methodology to use direct predictions from the final trained model [y = f(X1)] for these data points. This revision aligns with machine learning best practices, as the final data product should utilize predictions from the complete trained model rather than intermediate cross-validation results. The cross-validation metrics remain valuable for model evaluation during the development phase, while the final product benefits from the full model trained on the entire training dataset. For uncertainty quantification, we maintain the use of validation set RMSE, as it aligns well with 10-fold cross-validation results and provides a more comprehensive assessment of model uncertainty. Noted that this revision did not essentially change the results of this work.

**All revisions are highlighted in red in the manuscript and are detailed in this response letter.**

**R2C1.** One edit I have for the paper regards the interchangeable use of  $fCO_2$  and  $pCO_2$ . In section 2.1 the authors mention that "both are commonly used in oceanographic studies", which is accurate. However, they are not interchangeably used. At the end of Section 2.2 an equation to convert  $pCO_2$  to  $fCO_2$  is provided, but it's unclear at what point this conversion is made. Figure 2 shows values in  $fCO_2$ , but the rest of the figures use  $pCO_2$ . I recommend a clear statement about conversion with the introduction of  $fCO_2$ , as well as consistency in the figures (I would convert Figure 2 to showing  $pCO_2$  or at least have a clear statement on the conversion and reason for  $fCO_2$  presentation in the figure caption).

**Response:** We thank the reviewer for highlighting this important distinction between  $fCO_2$  and  $pCO_2$ . Throughout the main text, we primarily focused on  $pCO_2$  comparisons, as this parameter was directly provided by other products. We discovered an error, which actually shows  $pCO_2$  values in our Python codes but was incorrectly written as  $fCO_2$  in both the color bar and caption for Figure 2. We have now corrected these labels and added a clarifying sentence to the caption. The modified Figure 2 is attached for reference.

Figure 2. Spatial distribution of sea surface  $pCO_2$  observations from SOCAT database (version 2023) in the NAACOM across four seasons from 1993 to 2021. Grid samples with data were counted by season: (a) Spring (March to May), (b) Summer (June to August), (c) Fall (September to November), and (d) Winter (December to February). The study region is divided into northern (blue box) and southern (red box) areas at approximately 41.5°N (Cape Cod). The number and percentage of grid samples are indicated for each region per season. Color scale represents  $pCO_2$  values in µatm. Higher

sampling density is evident in the southern area. Winter shows the lowest overall sampling coverage. Note that the SOCAT database provides quality-controlled  $fCO_2$  measurements as the default parameter, which are subsequently converted to  $pCO_2$  using Eq. (2).

Regarding the usage of  $fCO_2$  in our study, because SOCAT reports  $fCO_2$  directly rather than  $pCO_2$ , we specifically use it as the label during model training. Based on our previous work on carbonate dynamic in the Gulf of Mexico (Wu et al., 2024), we decided to provide both reconstructed  $fCO_2$  and  $pCO_2$  in our final product. The model trains and outputs  $fCO_2$  directly, and the predicted  $fCO_2$  values are subsequently converted to  $pCO_2$  using OISST data. This process was briefly mentioned in the last paragraph of Section 2.2:

"Finally, the trained model is applied to all satellite and reanalysis data to generate the final gap-free reconstructed  $fCO_2$  data. As most products reported seawater  $CO_2$  concentration as  $pCO_2$ , our final product reports both  $fCO_2$  and  $pCO_2$ , with the  $fCO_2$  values being converted to  $pCO_2$  using the following equation (Takahashi et al., 2019)."

To further clarify this, we have made the following modifications:

- 1) We updated Figure 3 to show that the model first outputs  $3D fCO_2$ , which is then converted to  $pCO_2$
- 2) A clarifying sentence was added at the beginning of Section 2.2 (Model design, line 148): "During the model development phase, *f*CO2 measurements served as training labels for the machine learning algorithm."
- 3) We modified the last paragraph of Section 2.2 to emphasize that  $fCO_2$  was used in the model development phases before conversion to  $pCO_2$

The revised Figure 3 is attached for reference."

---

## Author Response (AR1)

**Review of Wu et al. (essd-2024-309)**

**General comments**

Wu et al. describe a new data product that reconstructs sea surface $pCO_2$ in the North American Atlantic coastal ocean margin over nearly thirty years. The authors rely on the gridded Surface Ocean $CO_2$ Atlas (SOCAT) dataset as the baseline observations for this data product, and they reconstruct $pCO_2$ using a two-step random forest regression (RFR) + linear regression (LR) approach. They find that their data product (ReCAD-NAACOM-$pCO_2$) effectively captures coastal features and variability along the North Atlantic coastal margin. The authors report a region-wide $R^2$ of 0.83 and RMSE of 18.64 µatm in comparison to observations. Overall, this manuscript describes a useful product that has value for those engaged in studies of ocean acidification and air-sea $CO_2$ flux in the region. There are some areas, however, where more detailed explanations and thorough analyses would make this a stronger contribution.

> **General responses:** We sincerely appreciate the reviewer's recognition of the value of our work. We have carefully addressed the reviewer's suggestions for strengthening our contribution through more detailed explanations and thorough analyses. The manuscript has been extensively revised to incorporate feedback from both reviewers. Major improvements include:
>
> 1. Clarified the usage and distinction between $fCO_2$ and $pCO_2$ throughout different sections of the manuscript
> 2. Enhanced the presentation and explanation of Mean Bias Error (MBE) in Section 3.2 and Fig. 5 to avoid potential confusion
> 3. Substantially revised Section 3.3 to better explain why product-estimated $pCO_2$ and SOCAT observations show discrepancies in the northern areas, attributing these differences to limited observational coverage
> 4. Restructured Section 3.4 and Fig. 7 to clearly differentiate between previously documented regional variations and newly identified phenomena revealed by our product
>
> Additionally, we have made an important revision regarding model evaluation. In our previous version, we reported the model outputs for the training dataset (80% of X1) using results from 10-fold cross-validation. We have now updated our methodology to use direct predictions from the final trained model [y = $f$(X1)] for these data points. This revision aligns with machine learning best practices, as the final data product should utilize predictions from the complete trained model rather than intermediate cross-validation results. The cross-validation metrics remain valuable for model evaluation during the development phase, while the final product benefits from the full model trained on the entire training dataset. For uncertainty quantification, we maintain the use of validation set RMSE, as it aligns well with 10-fold cross-validation results and provides a more comprehensive assessment of model uncertainty. Noted that this revision did not essentially change the results of this work.
>
> **All revisions are highlighted in red in the manuscript and are detailed in this response letter.**

**R1C1.** The strategy of adjusting RFR estimates with an LR is a unique and straightforward way to mitigate possible biases in the RFR estimates. However, this aspect of the methodology could use more explanation, in particular with respect to why this correction might be needed and how it improves the product ReCAD-NAACOM-$p$CO$_2$ relative to not implementing the LR step. If, as indicated, the LR serves to "mitigate potential systematic biases in RFR-derived $f$CO$_2$ values [that] arise from spatiotemporal heterogeneities in the SOCAT observational dataset", I envision a figure like Fig. 4c before and after applying the LR would emphasize the added value of this methodological step.

**Response:** We thank the reviewer for this constructive suggestion. To demonstrate the value of the LR calibration step, we have added two new figures comparing the performance before and after LR calibration across six sub-regions in **Appendix A**. These figures show both monthly climatology and $p$CO$_2$ trends. While the LR calibration yields modest improvements in monthly climatology, it significantly enhances the representation of monthly $p$CO$_2$ anomalies (deseasonalized), as evidenced by improved R² values and reduced RMSE. We have incorporated these findings into Results Section 3.1, lines 271-277:

"Our product employs a two-step RFR+LR algorithm to retrieve $p$CO$_2$. The initial RFR step accurately captures most seasonal and decadal $p$CO$_2$ variations across all six sub-regions (**Appendix A**). When comparing only at matching grid cells where SOCAT measurements are available, the differences ($N = 12$) in monthly mean climatology between SOCAT and RFR-derived $p$CO$_2$ are less than 2 µatm on average with standard deviations below 5 µatm across all sub-regions (**Fig. A1**). However, the RFR-derived $p$CO$_2$ shows lower accuracy in capturing long-term $p$CO$_2$ changes in the GoMe and SAB. The subsequent LR calibration improves the performance significantly: R² values increase from 0.69 to 0.81 in the GoMe and from 0.83 to 0.93 in the SAB, while RMSE decreases from 12.43 to 10.51 µatm in the GoMe and from 10.83 to 8.12 µatm in the SAB (**Fig. A2**)."

**R1C2.** I find the analysis presented in Section 3.3 to be somewhat lacking. While the similarities in large-scale climatological patterns between the raw observations and ReCAD-NAACOM-$p$CO$_2$ is encouraging, more interesting is where, when, and why the two datasets differ, and how those differences speak to the value added by the gap-filled product. In particular, I see much higher wintertime $p$CO$_2$ in the observations compared to the product in the northern region in Fig. 6. Is this result due to preferential observational coverage of high-$p$CO$_2$ areas in that season, as potentially indicated by Fig. 2d? This type of analysis I think is more interesting to readers, and more effective at communicating the utility of the new product.

**Response:** We thank the reviewer for this constructive and valuable suggestion. Following both reviewers' comments, we have expanded Section 3.3 to include a more detailed analysis of the differences between observations and our gap-filled product, particularly focusing on regional and seasonal variations. We have added new discussions about the sampling limitations in the northern regions and how our product addresses these gaps. For convenience, we attached the revisions below (lines 339-353):

" One of the primary objectives of this product is to capture the seasonal cycle of $p$CO$_2$ across the NAACOM region. **Figure 6** showcases the applicability of the product in capturing the $p$CO$_2$ seasonal cycles across the southern and northern areas of

NAACOM (red and blue boxes in **Fig. 2**). The comparison of monthly climatologies between the gap-filled product and SOCAT observations reveals strong agreement in the southern regions, despite of the coverage difference, with product-estimated monthly means being only $3.05 \pm 5.60$ µatm higher than SOCAT (**Fig. 6a**), suggesting that our product effectively captures the seasonal cycle where data are abundant.

In the northern regions where SOCAT data are sparse, the gap-filling ability of the product is also well demonstrated. In the northern region, the area-average monthly $pCO_2$ climatology calculated from the continuous reconstructed product are $22 \pm 11.12$ µatm lower than SOCAT observations, which can be attributed to limited observational coverage in this area. This area is characterized by sparse sampling, with observation density approximately 50% lower than in the southern regions (**Fig. 2**) due to the smaller area and limited cruise coverage. For instance, the GStL region only has one summer cruise in SOCAT database (**Fig. 2b**), and the SS and GoMe have particularly sparse winter observations (**Fig. 2d**). The higher latitudes typically exhibit larger seasonal amplitudes in $pCO_2$, making the limited sampling from SOCAT particularly problematic for accurate characterization. Our gap-free product provides comprehensive spatial and temporal coverage, enabling more robust analysis of $pCO_2$ patterns and variability in these historically under-sampled regions."

**R1C3.** The comparisons to global products detailed in Section 3.4 would benefit from some quantitative results to be presented alongside the qualitative interpretation of the annual mean climatological figures. The authors assert, for example, that compared to ULB_SOMFFN_coastal_v2 "the ReCAD-NAACOM-$pCO_2$ product exhibits closer values to the observations", but provide no evidence outside visual inspection of Fig. 7. Instead, by comparing (for instance) the average and RMSE of differences between the gridded SOCAT observations and corresponding values from the products within specific regions, the authors could more clearly emphasize the level of improvement provided by ReCAD-NAACOM-$pCO_2$.

**Response:** We thank the reviewer for catching this. Regarding Figure 7 in Section 3.4, our previous work (Wu et al., 2024) indicated that existing $pCO_2$ products did not adequately meet our requirements for regional analysis. Therefore, the objective of Fig.7 is to show the capability of this product in capturing these regional variations.

We agreed that Section 3.4 could be more quantitative and revised the relative descriptions. The revised section now presents two distinct components: (1) validation of previously confirmed regional variations, and (2) discussion of novel patterns revealed by our product that warrant future investigation. The expanded Section 3.4 is provided below (lines 380-398):

"The ReCAD-NAACOM-$pCO_2$ product demonstrates superior alignment with SOCAT observations in capturing these regional features that have been reported in previous observation-based studies (**Fig. 7b**) ….

In addition to these previously documented regional variations, our product reveals several notable features not previously captured by observations or other existing

products. For instance, the GoMe displays intermediate $pCO_2$ levels around 380 μatm, distinctly higher than surrounding waters at comparable latitudes, a feature previously documented by a multiple linear regression reconstructed $pCO_2$ product (Signorini et al., 2013) and five-year (2004-2009) mooring and cruise data (Vandemark et al., 2011) but contradict to another two studies based on numerical models (Cahill et al., 2016; Rutherford et al., 2021). In the southern GStL (S.GStL, box 4 in **Fig. 7**), $pCO_2$ values are slightly higher compared to adjacent waters at similar latitudes, aligning with high nutrient concentrations typically observed in these river-influenced waters (Lavoie et al., 2021). These regional patterns could not be completely captured by the global products (**Fig. 7c and 7d**). Ability of the ReCAD-NAACOM-$pCO_2$ product in resolving such regional features demonstrates its potential value for investigating coastal carbon dynamics and their responses to local and regional forcing factors in the NAACOM."

**Additional line-specific comments are provided below.**

**Specific comments**

**R1C4.** 49: Aren't the $f$CO$_2$ data included in SOCAT from these cruises exclusively from underway measurements (not discrete)? In which case, perhaps this sentence should read "Underway measurements from these research cruises, combined with underway measurements from volunteer observing ships and buoy observations, …" or something to that effect.

> **Response:** We appreciate the reviewer's attention to detail regarding the types of measurements included in our study. In response, we have modified the sentence to read (lines 51-52):
>
> "Underway measurements from these cruises, combined with underway measurements from volunteer observing ships and buoy, are quality-controlled and compiled in the Surface Ocean CO$_2$ Atlas (SOCAT) database (Bakker et al., 2016),"

**R1C5.** Figure 1: The regional labels might be better displayed in orange rather than red. As they are now, one might understandably associate the red labels with the red 200m isobath, which can be confusing.

> **Response:** We thank the reviewer for this helpful suggestion. We have changed the regional labels from red to orange to avoid confusion with the 200m isobath. Additional revisions to the figure have also been made based on suggestions from another reviewer. The modified Figure 1 is attached for reference:

[Figure]

**Figure 1. Topography (in meters) and large-scale circulation along the North American Atlantic Coastal Ocean Margin (NAACOM).** The study region, defined as coastal areas extending 400 km offshore, is indicated by blue shading. The thick black line is the 200 m isobath, which roughly marks the shelf break and typically defines the continental shelf boundary. The Gulf Stream (thick red dashed line with an arrow) flows northward along the east coast of the United States before veering eastward into the open Atlantic Ocean around Cape Hatteras. The Labrador Current (thick light blue dashed line with an arrow) flows southward along the east coast of Canada before meeting the Gulf Stream. Following Fennel et al. (2019), the study region is divided into six sub-regions by straight orange lines: the Gulf of Mexico (GoMx), South Atlantic Bight (SAB), Mid-Atlantic Bight (MAB), Gulf of Maine (GoMe), Scotian Shelf (SS), and Gulf of St. Lawrence and Grand Banks (GStL&GB). Dashed contour lines indicate bathymetric depths of 50 m and 100 m on the shelf (from coastline to 200 m isobath), and 1000 m, 2000 m, 3000 m, and 4000 m from the shelf break to the open ocean.

**R1C6.** 75–77: These two sentences say essentially the same thing and could be combined.

**Response:** We thank the reviewer for catching this redundancy. We have removed the second sentence.

**R1C7.** 105: The word "enhanced" suggests a comparison for the spatial, seasonal, and decadal variability. It should be mentioned here that the capability of the product at resolving these variations is enhanced in reference to some other dataset. Global products? The gridded SOCAT observations?

**Response:** We agree that clarification is needed to specify the reference point for our product's enhanced capabilities. In response, we have modified the original sentence to read (line 108):

"… enhanced capability in resolving spatial variations and capturing the seasonal cycle and decadal trends of $pCO_2$ better than those of the global products across different sub-regions along the NAACOM."

**R1C8.** 109: I find the "ground-truth data" terminology to be somewhat misleading. Ground-truth suggests data that is used to evaluate some model or remote-sensing measurement, but here the data is used not only as a ground-truth but also for training the model itself. Perhaps something like "observational data", "model training data", etc. might be more appropriate.

> **Response:** We appreciate the reviewer's suggestion regarding proper use of terminology. We initially adopted the term 'ground-truth data' from remote sensing and machine learning studies. However, we agree that in the context of our oceanographic research, "observational data" is a more appropriate and precise term. To address this, we have replaced three instances of "ground-truth data" with "observational data" in the manuscript.

**R1C9.** 118: Sampling density also looks to be particularly low in the western Gulf of Mexico.

> **Response:** We fully concur with the reviewer's observations. Indeed, the sampling density is also low in the western and southern Gulf of Mexico, as we previously noted in our Gulf of Mexico publication (Wu et al., 2024). We have modified the original sentence as follows (lines 121-122):
>
> "Observational data show lower sampling density in the areas north of Cape Cod and western and southern GoMx (blue box in **Fig. 2**)."

**R1C10.** Figure 3: My understanding is that the "Model" (light orange box with curved sides) is the same that is applied to all satellite and reanalysis data to construct the gridded product. As such, I'd recommend some modification to this flow chart. The arrow from "Model" to "Predictive model" is confusing if those two items are indeed the same.

> **Response:** We sincerely appreciate the reviewer's comments regarding the clarity of our original flowchart. We acknowledge that the initial representation could have been clearer, and we have made three modifications to address this concern:
>
> 1) We have made the two "models" the same in the flowchart.
>
> 2) We have added a sentence in the caption explicitly stating that 'The two models in the orange boxes are identical.' to avoid any potential confusion.
>
> 3) Following another reviewer's valuable suggestion, we have included the model outputs in the flowchart. This addition clarifies the sequential nature of our approach: our machine learning model first outputs $f\mathrm{CO}_{2\mathrm{sea}}$, which is then converted to $p\mathrm{CO}_{2\mathrm{sea}}$ using OISST data.
>
> The revised flowchart with caption is attached:

[Figure]

**Figure 3. A flowchart of the two-step machine learning regression model for generating the reconstructed $p$CO₂ product.** Grey boxes represent the input and output datasets, blue boxes illustrate the model training, validation testing, and independent test processes, and orange boxes represent the final trained model for predicting the reconstructed product. The two models in the orange boxes are identical. The training data, consisting of paired input variables (lon, lat, month, sea surface temperature (SST), sea surface salinity (SSS), sea surface height (SSH), and atmospheric $p$CO₂ ($p$CO₂air) and corresponding sea surface $f$CO₂ ($f$CO₂sea) labels), is divided into two sets: X1 (1993-2003 and 2006-2021) and X2 (2004-2005). X1 is further randomly divided into subsets for model training set (80%) and validation set (20%). The predictive model combines a random forest regression (RFR) and a linear regression (LR) algorithm. The trained and validated regression model is then applied to all satellite and reanalysis data (without gaps) to generate the 3D reconstructed $f$CO₂sea product, which was then converted to $p$CO₂sea with satellite SST data.

**R1C11.** 167–168: More explanation should be given here on exactly how the validation set is used to evaluate the model performance.

> **Response:** We appreciate the reviewer's suggestion for clarification regarding the validation set's role in model evaluation. We have expanded our explanation as follows (lines 176-182):
>
> "The validation set, comprising 20% of X1 randomly sampled from 1993-2003 and 2006-2021, serves as a critical monitoring step for model evaluation. This subset plays two key roles: first, it tests hyperparameter tuning by providing independent performance metrics on unseen data, and second, it helps detect potential overfitting by monitoring the divergence between training and validation performance. While the validation set itself cannot prevent overfitting, it enables detection of overfitting patterns when performance of the model improves on training data but deteriorates on validation data. Through this continuous evaluation process, the validation set ensures more robust model development and helps achieving better generalization capabilities."

**R1C12.** 171–172: This sentence is somewhat unclear.

**Response:** We appreciate the reviewer's suggestion for clarification. We have modified our explanation as follows (lines 183-186):

"The independent test set (X2), covering the years 2004-2005, serves as a critical evaluation period specifically designed to assess reliability of the model in predicting values for years that were completely excluded from both training and validation phases. Because we intentionally withhold these two years from model development, this approach directly tests capability of the model in generating reliable predictions and fill temporal data gaps for periods without observational data."

**R1C13.** 178: How is month treated in the model training? If you're only using 1–12 for the months of the year, there will be an unintended effect whereby months that should be treated as similar (e.g., January vs. December) will be treated as extremely different (1 vs. 12). See Sauzède et al. (2015) or Gregor et al. (2018) for information about transforming cyclical predictors using sine and cosine functions.

**Response:** We used numerical values 1-12 to represent months in our algorithm. We agree with the reviewer's comment regarding the treatment of monthly data and the potential artificial discontinuity. Following this suggestion, we conducted additional analyses by implementing the suggested sinusoidal transformation $[\sin(month/12 * 2\pi)]$ and reran our complete modeling framework. The comparisons are shown in the table bellow. Our analysis revealed that this modification resulted in minimal differences in the model output matrix, with variations comparable to those stemming from random sampling algorithms.

| Months | $R^2$ | RMSE (µatm) | MAE (µatm) | MBE (µatm) |
|---|---|---|---|---|
| 1-12 | 0.92 | 12.70 | 7.55 | 0.13 |
| $\sin(month/12 * 2\pi)$ | 0.90 | 14.59 | 8.70 | -0.17 |

These tests suggest that the seasonal cycle information in our study region is largely captured by other variables (SST, SSS, SSH, and $p\mathrm{CO}_{2air}$), which inherently contain seasonal patterns. Based on these findings, we conclude that the minimal impact of the monthly representation method indicates our current conclusions remain robust.

Given no improvements in model performance, we maintained the original methodology to preserve consistency in the manuscript's statistical analyses, as the Mente Carlo simulation. However, we acknowledge that implementing proper cyclical variable treatment is theoretically more appropriate. In our ongoing development of version 2 of this product, which includes reconstructed SSS fields for the entire Pacific and Atlantic margins (currently under validation), we plan to implement the sinusoidal transformation for monthly variables.

**R1C14.** 206: I believe P represents the total atmospheric pressure, not the "$CO_2$ atmospheric pressure".

**Response:** We sincerely appreciate the reviewer's careful reading and attention to detail. Yes, the original text contained a typo. We have corrected this inaccuracy and revised

the text as follows:

"where $P$ is the total atmospheric pressure on the sea surface, …"

**R1C15.** 315–324: I'm not sure this discussion is very valuable because the general features discussed here are evident in the product but also in the observations themselves. It might be more effective to discuss the seasonal cycle features in the product as they relate to the observations; what information is added by the gap-filled product?

    **Response:** We thank the reviewer for this constructive comment. We have revised this paragraph to better highlight how our gap-free product enhances our understanding of seasonal cycles beyond what is visible in the raw observations. The revised discussion now quantifies the agreement between our reconstructed product and SOCAT observations, and explains regional differences in their monthly climatologies:

[Figure]

**Figure 6. Monthly mean climatology of $p$CO$_2$ in the southern and northern areas of the NAACOM from 1993 to 2021.** Sub-regions are **(a)** southern areas, the red box in Fig. 2, and **(b)** northern areas, the blue box in Fig. 2. Two data representations are shown: (1) SOCAT observations (black curves), which may be influenced by missing data; and (2) the complete gap-filled product output (red curves). Error bars denote one standard deviation of the monthly mean climatology of $p$CO$_2$. Numbers indicate the mean difference (± one standard deviation) between monthly climatological $p$CO$_2$ calculated from the two sources, with positive values indicating higher product estimates compared to SOCAT observations. The x-axis shows months (1-12, where 1 represents January), and the y-axis shows $p$CO$_2$ in µatm.

"One of the primary objectives of this product is to capture the seasonal cycle of $p$CO$_2$ across the NAACOM region. **Figure 6** showcases the applicability of the product in capturing the $p$CO$_2$ seasonal cycles across the southern and northern areas of NAACOM (red and blue boxes in **Fig. 2**). The comparison of monthly climatologies between the gap-filled product and SOCAT observations reveals strong agreement in the southern regions, despite of the coverage difference, with product-estimated monthly means being only 3.05 ± 5.60 µatm higher than SOCAT (**Fig. 6a**), suggesting that our product effectively captures the seasonal cycle where data are abundant.

In the northern regions where SOCAT data are sparse, the gap-filling ability of the product is also well demonstrated. In the northern region, the area-average monthly $p$CO$_2$ climatology calculated from the continuous reconstructed product are 22 ± 11.12 µatm lower than SOCAT observations, which can be attributed to limited observational coverage in this area. This area is characterized by sparse sampling, with observation density approximately 50% lower than in the southern regions (**Fig. 2**) due to the

smaller area and limited cruise coverage. For instance, the GStL region only has one summer cruise in SOCAT database (**Fig. 2b**), and the SS and GoMe have particularly sparse winter observations (**Fig. 2d**). The higher latitudes typically exhibit larger seasonal amplitudes in $p$CO$_2$, making the limited sampling from SOCAT particularly problematic for accurate characterization. Our gap-free product provides comprehensive spatial and temporal coverage, enabling more robust analysis of $p$CO$_2$ patterns and variability in these historically under-sampled regions."

**R1C16.** 416: It should be clarified here that this uncertainty value for the North American Pacific Coastal Ocean Margin is specific to areas within 100km of the coastline and the uncertainty provided for ReCAD-NAACOM-$p$CO$_2$ is for areas within 400km.

> **Response:** We appreciate the reviewer's careful attention to detail. We have revised the original sentence to make it more precise (lines 454-456):

> "Despite this conservative method, our calculated uncertainty for the Atlantic margins is comparable to the 43.4 μatm reported by Sharp et al. (2022) for areas within 100 km of the North American Pacific margins. suggesting a good product performance of our product."

**Technical corrections**

**R1C17.** 148: Should be "arising" or "that arise"

> **Response:** We appreciate the reviewer's careful attention to detail. We have modified the original phrase "... arise from ..." to "... arising from ...". The revised sentence now reads (lines 154-157):

> "**…** while the LR model is subsequently applied to mitigate potential systematic biases in RFR-derived $f$CO$_2$ values arising from spatiotemporal heterogeneities in the SOCAT observational dataset **(Fig. 2)**."

**R1C18.** 424: Recommend changing wording here: "the performance…reduced" is somewhat awkward

> **Response:** We thanks for pointing out this and revised the original sentence to (line 466):

> "Furthermore, during the independent validation phase, the accuracy of the model predicted values reduced in the GoMe …"

**References**

Wu, Z., Wang, H., Liao, E., Hu, C., Edwing, K., Yan, X.-H., & Cai, W.-J. (2024). Air-sea CO2 flux in the Gulf of Mexico from observations and multiple machine-learning data products. *Progress in Oceanography*, *223*, 103244. https://doi.org/10.1016/j.pocean.2024.103244

**Response letter to Reviewer#2**

**Overview**

The authors are presenting a new, regional $p$CO$_2$-product specifically designed for the North American Atlantic coastal region that provides monthly $p$CO$_2$ at a .25-degree spatial resolution from 1993-2021. The product uses integrated random forest and linear regression methods incorporating observational products in order to generate their monthly reconstructed $p$CO$_2$-product. This allows for analysis of regional, seasonal, and yearly trends in addition to an uncertainty calculation. The authors find through validation that their product provides high accuracy, improving public access for more precise, higher resolution coastal carbon dynamics in the NAACOM region.

There is great need for products like these to be publicly accessible, and this contributes an important resource to the scientific community. The authors do a nice job of introducing the field, the data gaps, and where this product can contribute to those gaps. I do recommend for publication, following a few edits as outlined below.

> **General responses:** We sincerely appreciate the reviewer's recognition of the value of our work. We have carefully addressed the reviewer's suggestions for strengthening our contribution through more detailed explanations and thorough analyses. The manuscript has been extensively revised to incorporate feedback from both reviewers. Major improvements include:
>
> 5. Clarified the usage and distinction between $f$CO$_2$ and $p$CO$_2$ throughout different sections of the manuscript
> 6. Enhanced the presentation and explanation of Mean Bias Error (MBE) in Section 3.2 and Fig. 5 to avoid potential confusion
> 7. Substantially revised Section 3.3 to better explain why product-estimated $p$CO$_2$ and SOCAT observations show discrepancies in the northern areas, attributing these differences to limited observational coverage
> 8. Restructured Section 3.4 and Fig. 7 to clearly differentiate between previously documented regional variations and newly identified phenomena revealed by our product
>
> Additionally, we have made an important revision regarding model evaluation. In our previous version, we reported the model outputs for the training dataset (80% of X1) using results from 10-fold cross-validation. We have now updated our methodology to use direct predictions from the final trained model [y = $f$(X1)] for these data points. This revision aligns with machine learning best practices, as the final data product should utilize predictions from the complete trained model rather than intermediate cross-validation results. The cross-validation metrics remain valuable for model evaluation during the development phase, while the final product benefits from the full model trained on the entire training dataset. For uncertainty quantification, we maintain the use of validation set RMSE, as it aligns well with 10-fold cross-validation results and provides a more comprehensive assessment of model uncertainty. Noted that this revision did not essentially change the results of this work.

**All revisions are highlighted in red in the manuscript and are detailed in this response letter.**

**R2C19.** One edit I have for the paper regards the interchangeable use of *f*CO$_2$ and *p*CO$_2$. In section 2.1 the authors mention that "both are commonly used in oceanographic studies", which is accurate. However, they are not interchangeably used. At the end of Section 2.2 an equation to convert *p*CO$_2$ to *f*CO$_2$ is provided, but it's unclear at what point this conversion is made. Figure 2 shows values in *f*CO$_2$, but the rest of the figures use *p*CO$_2$. I recommend a clear statement about conversion with the introduction of *f*CO$_2$, as well as consistency in the figures (I would convert Figure 2 to showing *p*CO$_2$ or at least have a clear statement on the conversion and reason for *f*CO$_2$ presentation in the figure caption).

> **Response:** We thank the reviewer for highlighting this important distinction between *f*CO$_2$ and *p*CO$_2$. Throughout the main text, we primarily focused on *p*CO$_2$ comparisons, as this parameter was directly provided by other products. We discovered an error, which actually shows *p*CO$_2$ values in our Python codes but was incorrectly written as *f*CO$_2$ in both the color bar and caption for Figure 2. We have now corrected these labels and added a clarifying sentence to the caption. The modified Figure 2 is attached for reference.

[Figure]

**Figure 2. Spatial distribution of sea surface *p*CO$_2$ observations from SOCAT database (version 2023) in the NAACOM across four seasons from 1993 to 2021.** Grid samples with data were counted by season: **(a)** Spring (March to May), **(b)** Summer (June to August), **(c)** Fall (September to November), and **(d)** Winter (December to February). The study region is divided into northern (blue box) and southern (red box) areas at approximately 41.5°N (Cape Cod). The number and percentage of grid samples are indicated for each region per season. Color scale represents *p*CO$_2$ values in µatm. Higher

sampling density is evident in the southern area. Winter shows the lowest overall sampling coverage. Note that the SOCAT database provides quality-controlled $fCO_2$ measurements as the default parameter, which are subsequently converted to $pCO_2$ using Eq. (2).

Regarding the usage of $fCO_2$ in our study, because SOCAT reports $fCO_2$ directly rather than $pCO_2$, we specifically use it as the label during model training. Based on our previous work on carbonate dynamic in the Gulf of Mexico (Wu et al., 2024), we decided to provide both reconstructed $fCO_2$ and $pCO_2$ in our final product. The model trains and outputs $fCO_2$ directly, and the predicted $fCO_2$ values are subsequently converted to $pCO_2$ using OISST data. This process was briefly mentioned in the last paragraph of Section 2.2:

"Finally, the trained model is applied to all satellite and reanalysis data to generate the final gap-free reconstructed $fCO_2$ data. As most products reported seawater $CO_2$ concentration as $pCO_2$, our final product reports both $fCO_2$ and $pCO_2$, with the $fCO_2$ values being converted to $pCO_2$ using the following equation (Takahashi et al., 2019)."

To further clarify this, we have made the following modifications:

1) We updated Figure 3 to show that the model first outputs 3D $fCO_2$, which is then converted to $pCO_2$
2) A clarifying sentence was added at the beginning of Section 2.2 (Model design, line 148): "During the model development phase, $fCO_2$ measurements served as training labels for the machine learning algorithm."
3) We modified the last paragraph of Section 2.2 to emphasize that $fCO_2$ was used in the model development phases before conversion to $pCO_2$

The revised Figure 3 is attached for reference."

[Figure]

**Figure 3. A flowchart of the two-step machine learning regression model for generating the reconstructed $pCO_2$ product.** Grey boxes represent the input and output datasets, blue boxes illustrate the model training, validation testing, and independent test processes, and orange boxes represent the final trained model for predicting the reconstructed product. The two models in the orange boxes are

identical. The training data, consisting of paired input variables (lon, lat, month, sea surface temperature (SST), sea surface salinity (SSS), sea surface height (SSH), and atmospheric $pCO_2$ ($pCO_{2air}$) and corresponding sea surface $fCO_2$ ($fCO_{2sea}$) labels), is divided into two sets: X1 (1993-2003 and 2006-2021) and X2 (2004-2005). X1 is further randomly divided into subsets for model training set (80%) and validation set (20%). The predictive model combines a random forest regression (RFR) and a linear regression (LR) algorithm. The trained and validated regression model is then applied to all satellite and reanalysis data (without gaps) to generate the 3D reconstructed $fCO_{2sea}$ product, which was then converted to $pCO_{2sea}$ with satellite SST data.

**R2C20.** My second edit has to do with the calculation of uncertainty. There is a calculation representing the inputs ($u_{inputs}$), but some of the products are in the original resolution and others are linearly interpolated to that resolution. Does the interpolation introduce more error, and is this taken into account?

> **Response:** We appreciate the reviewer's inquiry regarding our input variables. Our model uses sea surface height (SSH), sea surface temperature (SST), sea surface salinity (SSS), and atmospheric $pCO_2$ as input variables. To clarify, 1) both SSH and SST are at 0.25° resolution and monthly timescale, averaged from daily data. 2) Atmospheric $pCO_2$ has a very small spatial gradient when compare with the sea surface $pCO_2$. Thus, interpretation won't introduce additional uncertainty. In our previous studies, we found that even Mauna Loa $pCO_2$ data would closely approximate the $pCO_2$ observed at U.S. East Coast stations, as $pCO_2$ typically varies little globally (Wu et al., 2024).
>
> SSS is the variable most likely to introduce uncertainty due to interpolation. It's crucial as it reflects the interaction between terrestrial and open ocean waters. While SODA is a widely used and respected salinity product, its uncertainty in our study region is not well-defined. Therefore, we adopted a conservative approach: **we doubled the uncertainty reported in the SODA paper for our calculations.** To clarify this in our methods, we've added a sentence in the Method section (lines 210-211):
>
> "Noted that such interpolation could potentially introduce additional errors. We doubled the SSS uncertainty in the region, assuming this would encompass its true uncertainty (see **Appendix B**)."
>
> And in Appendix B, line 535:
>
> "Their analysis (their Fig. 8) indicates an RMSE exceeding 0.3 psu in the vicinity of our area of interest. In addition, interpolating the 0.5° SSS data to 0.25° resolution could potentially introduce more errors. To maintain a conservative approach in our uncertainty quantification, we doubled the uncertainty and adopted a value of 0.6 psu as the SSS uncertainty for our calculations."
>
> We acknowledge that SODA SSS may not be optimal in our outer study region. Our Monte Carlo simulations indicate that the SSS-induced uncertainty (around 4 µatm) is small compared to the final uncertainty around 20 µatm, but we agree that a more accurate SSS product would be preferable. To the best of our knowledge, currently, no

such reliable coastal SSS data are publicly available for this region. We are developing an improved SSS product for this area, which requires further evaluation. Once we are confident in its reliability, we plan to make it publicly available and incorporate it into ReCAD version 2.

**R2C21.** Additionally, I will note that I greatly appreciate the inclusion of an uncertainty calculation and the strength it lends to the product. I would have liked to see it highlighted a bit more in the rest of the paper results—some of the data could also be discussed with uncertainty included, rather than purely keeping the uncertainty in one section at the end of the paper. I think the addition of the uncertainty calculation makes this product stronger, and should be displayed as such.

> **Response:** We appreciate the reviewer raising this important point. While writing the manuscript, we deliberated but ultimately decided not to include error analysis for the monthly mean and regional average, to avoid confusion and maintain focus and conciseness. Please see **R2C10** for detailed explanation.

**R2C22.** Finally, Figure 5 and figure 7, and the associated discussions in 3.2 and 3.4, present some extremely interesting data. We compare some of the regional differences and the products effectiveness of capturing broader $pCO_2$ patterns across the North Atlantic coast. I would have loved to had this extrapolated on a little further, and perhaps seen more numbers broken down by region. We can visually look at the figures, but it's a little hard to assess and I think the paper would be strengthened by expanding this section a little more with increased quantitative results.

> **Response:** We thank the reviewer for this constructive suggestion. We have enhanced the quantitative presentation of our results in several ways. Figure 5 illustrates the spatial distribution of mean bias error (MBE), these values complement the subregional MBE statistics already provided in the last column of Table 2. To make this more clear, we have now incorporated the average MBE values for all six subregions directly within the figure. We have also expanded the discussion section with additional quantitative analysis of regional patterns. **Please see our response to R2C8 for further details of these modifications.**
>
> Regarding Figure 7 in Section 3.4, our previous work (Wu et al., 2024) indicated that existing $pCO_2$ products did not adequately meet our requirements for regional analysis. Therefore, one of our primary objectives of this work was to develop a product to capture these regional variations. The $pCO_2$ distribution shows a clear south-to-north decreasing gradient, with distinct regional patterns superimposed on this large-scale distribution. We selected specific regions where $pCO_2$ variations have been well-documented in previous studies, including elevated $pCO_2$ in the West Florida Shelf (WFS), and low $pCO_2$ in the Mississippi River estuary.
>
> Our product also reveals several interesting features, such as relatively higher $pCO_2$

values in the Gulf of Maine (GoMe, compared to surrounding waters) that potentially associated with local high-$p$CO$_2$ river estuary waters, and notable regional variations in northern areas. The northern regions are particularly interesting as previous studies have reported conflicting results, with some identifying these areas as CO$_2$ sinks and others as CO$_2$ sources. However, those spatial variations haven't been confidently confirmed by observations yet.

Following the reviewer's suggestion, we have expanded Section 3.4 to provide a more comprehensive analysis. The revised section now presents two distinct components: (1) validation of previously confirmed regional variations, and (2) discussion of novel patterns revealed by our product that warrant future investigation. The expanded Section 3.4 is provided below:

"The ReCAD-NAACOM-$p$CO$_2$ product demonstrates superior alignment with SOCAT observations in capturing these regional features that have been reported in previous observation-based studies (**Fig. 7b**) ….

In addition to these previously documented regional variations, our product reveals several notable features not previously captured by observations or other existing products. For instance, the GoMe displays intermediate $p$CO$_2$ levels around 380 μatm, distinctly higher than surrounding waters at comparable latitudes, a feature previously documented by a multiple linear regression reconstructed $p$CO$_2$ product (Signorini et al., 2013) and five-year (2004-2009) mooring and cruise data (Vandemark et al., 2011) but contradict to another two studies based on numerical models (Cahill et al., 2016; Rutherford et al., 2021). In the southern GStL (S.GStL, box 4 in **Fig. 7**), $p$CO$_2$ values are slightly higher compared to adjacent waters at similar latitudes, aligning with high nutrient concentrations typically observed in these river-influenced waters (Lavoie et al., 2021). These regional patterns could not be completely captured by the global products (**Fig. 7c and 7d**). Ability of the ReCAD-NAACOM-$p$CO$_2$ product in resolving such regional features demonstrates its potential value for investigating coastal carbon dynamics and their responses to local and regional forcing factors in the NAACOM."

We plan to use this data product to discuss further the spatial and season variability of all sub-regions in NAACOM and the decadal trends in the data-rich MAB and SAB sub-regions in our subsequent publications.

**Specific Comments:**

**R2C23.** Figure 1: I felt that the colors of this figure made it difficult to interpret. The way the lines were drawn made the topography difficult to see. For consistency with the other figures in the paper, I would suggest shifting the coastal contour line to being black. Then perhaps make the Gulf stream and Labrador current lines dotted or dashed lines (also, change Gulf Stream's color if you shift contour to black), so they don't block as much topography. I would match the labels of the regions to the lines denoting the regions, and finally increase the

deviation in the color scale. Right now, it's not very easy to tell a difference between 800-1000m, and similarly between 0-300m is all about the same tone.

**Response:** We thank the reviewer for these suggestions to improve the figure's clarity. We agree with the reviewer regarding the visualization challenges of marine terrain data. The bathymetry in our study area presents a particular challenge due to the flat topography on the shelf (0-200 m), but rapid changes on the slope from 200m-2000m depth. For the colors, while we initially attempted to maintain consistent colors for the 200m isobath across all figures, this proved challenging due to the different colormaps required for various figures. To address these concerns, we have made the following modifications to Figure 1:

1) Implemented a light blue background to represent areas within 400 km of the coast
2) Added dashed contour lines for shelf depths (-50 m, -100 m, -200 m) and slope depths (-1000 m, -2000 m, -3000 m, -4000 m)
3) Updated the 200 m isobath to a thick black line in all figures for consistency
4) Modified the Gulf Stream and Labrador Current indicators to dashed arrows, following the reviewer's suggestion. The Gulf Stream is colored red to represent warm current and the Labrador Current is colored blue to represent cold current. Both are semi-transparent to avoid obscuring the contour lines
5) Added a legend to help readers interpret the lines in the figure

We have also revised the figure caption accordingly. The modified Figure 1 is attached for reference:

[Figure]

**Figure 2. Topography (in meters) and large-scale circulation along the North American Atlantic Coastal Ocean Margin (NAACOM).** The study region, defined as coastal areas extending 400 km offshore, is indicated by blue shading. The thick black line is the 200 m isobath, which roughly marks the shelf break and typically defines the continental shelf boundary. The Gulf Stream (thick red dashed

line with an arrow) flows northward along the east coast of the United States before veering eastward into the open Atlantic Ocean around Cape Hatteras. The Labrador Current (thick light blue dashed line with an arrow) flows southward along the east coast of Canada before meeting the Gulf Stream. Following Fennel et al. (2019), the study region is divided into six sub-regions by straight orange lines: the Gulf of Mexico (GoMx), South Atlantic Bight (SAB), Mid-Atlantic Bight (MAB), Gulf of Maine (GoMe), Scotian Shelf (SS), and Gulf of St. Lawrence and Grand Banks (GStL&GB). Dashed contour lines indicate bathymetric depths of 50 m and 100 m on the shelf (from coastline to 200 m isobath), and 1000 m, 2000 m, 3000 m, and 4000 m from the shelf break to the open ocean.

**R2C24.** Equations: center the equations in the document

**Response**: We appreciate the reviewer's attention to detail regarding equation formatting. Upon re-examination of the ESSD templates, we have confirmed that left-aligned equations are indeed the default format specified in both the ESSD Microsoft Word and LaTeX templates. We have decided to maintain this alignment to adhere to the journal's standard formatting guidelines, unless otherwise instructed by the editorial team. We thank the reviewer again for their careful revision.

**R2C25.** Lines 183-184: The nature of this sentence is implying an interchangeable use of $f\mathrm{CO}_2$ and $p\mathrm{CO}_2$, which I don't think is accurate

**Response:** We appreciate the reviewer's suggestion and agree that clarification was needed. We have modified the sentence as follows (lines 197-198):

"$p\mathrm{CO}_{2\mathrm{air}}$ represents the atmospheric forcing on the air-sea $\mathrm{CO}_2$ exchange. Including $p\mathrm{CO}_{2\mathrm{air}}$ is essential for accurately assessing the decadal $p\mathrm{CO}_2$ trend. "

**R2C26.** Line 284+: Authors show an area-mean bias of +0.17, but with the regional breakdown and discussion, I'd be very curious how that bias varies by region. Can we see numbers for the other regions as well?

**Response:**

The values shown in this figure represent the Mean Bias Error (MBE). We have updated our methodology to use direct predictions from the final trained model [ $y = f(80\%$ of X1)] for the training dataset (80% of X1), replacing our previous approach that used 10-fold cross-validation outputs. Consequently, the overall MBE for the entire region has been updated to +0.13 ±12.70 µatm. The regional mean bias values are reported in the last column of Table 2. For clarity, we have now incorporated these bias errors with their standard deviations directly into Figure 5:

[Figure]

**Figure 5. Spatial distribution of mean bias error (MBE) between ReCAD-NAACOM-*p*CO₂ product and SOCAT observations across the NAACOM.** The MBE is calculated for each grid cell as the average difference between product estimates and SOCAT observations. Positive values (red) indicate product overestimation, while negative values (blue) indicate underestimation relative to SOCAT. Regional MBE values with one standard deviation are shown for each sub-region, corresponding to the values in the last column of Table 2. The overall bias error for the NAACOM is +0.13 ± 12.97 µatm. Following Fennel et al. (2019), the study region is divided into six sub-regions by straight orange lines: the Gulf of Mexico (GoMx), South Atlantic Bight (SAB), Mid-Atlantic Bight (MAB), Gulf of Maine (GoMe), Scotian Shelf (SS), and Gulf of St. Lawrence and Grand Banks (GStL&GB). The thick black line is the 200 m isobath, which roughly marks the shelf break and typically defines the continental shelf boundary.

We also added two sentences to describe the distribution of MBE more quantitatively in lines 326-329:

"… Regional MBE for different machine learning development phases (training, validation, and test sets) are detailed in Table 2. Despite these regional differences, MBE of both the validation set (-1.0 ~ 1.0 µatm) and independent test set (-4.5 ~ 7.5 µatm) demonstrate minimal values across sub-regions (Table 2), underscoring the product's effectiveness in capturing the broader *p*CO₂ patterns across the NAACOM."

**R2C27.** Figure 5: Similar edit suggestions to figure 1; update contour line to be black and match the colors of regional names to the lines denoting the regions

**Response:** We thank the reviewer for these suggestions regarding visual consistency. Following the reviewer's recommendations, we have modified Figure 5 to maintain consistency with Figure 1 by:

1) Updating the 200 m isobath to a thick black line
2) Matching the colors of regional boundary lines and their corresponding region

names

3) Revising the figure caption accordingly

The modified Figure 5 is attached in the previous comment.

**R2C28.** Figure 6: the error bar denotes one standard deviation of the monthly mean climatology, but didn't the authors also actually calculate a $pCO_2$ error? Why is that not included in any of the figures?

> **Response:** We appreciate the reviewer raising this important point. While writing the manuscript, we deliberated but ultimately decided not to include error analysis for the monthly mean and regional average, to avoid confusion and maintain focus and conciseness. The error metrics shown in our figures serve different purposes:
>
> 1) The error bars in Fig. 6 and the values after "±" sign in Fig. 7 represent standard deviations (σ), which characterize the natural variability of pCO2 - temporal variability across the 29-year period (1993-2021) in Fig. 6, and spatial variability in Fig. 7.
> 2) The uncertainties reported elsewhere in the manuscript and provided in the NetCDF file reflect the propagated errors of individual $pCO_2$ estimates in each grid cell.
> 3) While these grid-cell uncertainties can be used to calculate uncertainties for area and monthly averages (following methods detailed in Roobaert et al., 2024 and Landschützer et al., 2014), we have deliberately focused this manuscript on establishing the fundamental reliability of our product and decided not to include this part in this manuscript to avoid potential confusion.
>
> A comprehensive analysis of uncertainties in regional averages and their implications for $CO_2$ flux calculations will be presented in our forthcoming manuscript examining $pCO_2$ seasonal cycles, regional variability, and mean seawater $CO_2$ uptake in the NAACOM. To ensure proper use of our dataset, we have added guidance in lines 459-461:
>
> "… In addition, the uncertainties reported in this section and provided in the NetCDF file represent the propagated errors for individual $pCO_2$ values in each grid cell. Methods to calculate uncertainties in regional averages of $pCO_2$ or air-sea $CO_2$ fluxes over specific spatial and temporal domains are detailed in Roobaert et al. (2024) and Landschützer et al. (2014)."
>
> **Detailed methods for calculating the uncertainty of monthly or area means (which did not appear in the text):**
>
> The uncertainty of monthly or regional means ($\theta_{mean}$) can be expressed as:
>
> $$\theta_{mean} = \sqrt{\theta_{stderr}^2 + \theta_{pCO2}^2}$$
>
> where $\theta_{stderr}$ is the **standard error** of monthly or regional means, which is a function of the standard deviation ($\sigma_{pCO2}$) for each month and the number of samples ($N_1 = 29$

for monthly means, or $N_1$ = total grid cells in a specific region for regional average):

$$\theta_{stderr} = \frac{\sigma_{pCO2}}{\sqrt{N_1}}$$

$\theta_{pCO2}$ is the uncertainty of product-estimated area-average $pCO_2$ accumulated from each grid cell, which could be calculated following Landschützer et al. (2014) and Roobaert et al. (2024):

$$\theta_{pCO2} = \sqrt{\frac{u_{obs}^2}{N_2} + \frac{u_{grid}^2}{N_2} + \frac{u_{map}^2}{N_{eff}}}$$

where $u_{obs}$, $u_{grid}$, and $u_{map}$ represent observational uncertainties, gridded uncertainty, and mapping uncertainty, respectively, as defined in our manuscript. $N_2$ denotes the number of grid cells in each region. For $u_{map}$, the value of N is corrected to effective sample size $N_{eff}$ as individual errors from each grid cell are not spatially independent, which could also be calculated with a Monte Carlo simulation (Landschützer et al., 2018).

We will report the **uncertainty of monthly or regional means** together with our scientific question--what controls regional and seasonal variability--in a separate publication.

**R2C29.** Line 415: The statement "This uncertainty is deemed reasonable" confused me. Deemed reasonable by who? What metrics are being used? "reasonable" is a very vague term.

> **Response:** We appreciate the reviewer's observation and agree that our original expression lacked precision. We have revised our approach explanation and comparison as follows in lines 453-456:
>
> "Our uncertainty estimation employs a conservative estimation, using maximum values at calculation step. This approach likely overestimates the true uncertainty. Despite this conservative method, our calculated uncertainty for the Atlantic margins is comparable to the 43.4 µatm reported by Sharp et al. (2022) for areas within 100 km of the North American Pacific margins. suggesting a good product performance of our product"

**R2C30.** Acknowledgments: Make sure to include the SOCAT statement from the website that they ask you to include when you use their product ("The Surface Ocean $CO_2$ Atlas (SOCAT) is an international effort, endorsed by the International Ocean Carbon Coordination Project (IOCCP), the Surface Ocean Lower Atmosphere Study (SOLAS) and the Integrated Marine Biosphere Research (IMBeR) program, to deliver a uniformly quality-controlled surface ocean $CO_2$ database. The many researchers and funding agencies responsible for the collection of data and quality control are thanked for their contributions to SOCAT.")

> **Response:** We appreciate the reviewer's suggestion and totally agree on the importance of acknowledging the scientific community's contributions to the data publicly, especially for the SOCAT effort. We have revised our acknowledgments section

accordingly and have also carefully reviewed the websites of other datasets used in this work to ensure comprehensive recognition. The updated acknowledgments now read as follows:

"The authors acknowledge the National Oceanic and Atmospheric Administration (NOAA) for providing the OISST data, the University of Maryland Ocean Climate Laboratory for the SODA dataset, and the European Union Copernicus Marine Service Information (CMEMS) for the SSH data. We also express our gratitude to the scientific community for sharing their observational carbonate data in the SOCAT effort. The SOCAT is an international effort, endorsed by the International Ocean Carbon Coordination Project (IOCCP), the Surface Ocean Lower Atmosphere Study (SOLAS) and the Integrated Marine Biosphere Research (IMBeR) program, to deliver a uniformly quality-controlled surface ocean $CO_2$ database. The many researchers and funding agencies responsible for the collection of data and quality control are thanked for their contributions to SOCAT.

This work is part of Zelun Wu's Ph.D. Dissertation under the University of Delaware-Xiamen University Dual Degree Program in Oceanography."

**References**

Cahill, B., Wilkin, J., Fennel, K., Vandemark, D., & Friedrichs, M. A. M. (2016). Interannual and seasonal variabilities in air-sea $CO_2$ fluxes along the U.S. eastern continental shelf and their sensitivity to increasing air temperatures and variable winds: U.S. East Coast Shelf Air-Sea $CO_2$ Fluxes. *Journal of Geophysical Research: Biogeosciences*, *121*(2), 295–311. https://doi.org/10.1002/2015JG002939

Fennel, K., Alin, S., Barbero, L., Evans, W., Bourgeois, T., Cooley, S., et al. (2019). Carbon cycling in the North American coastal ocean: a synthesis. *Biogeosciences*, *16*(6), 1281–1304. https://doi.org/10.5194/bg-16-1281-2019

Landschützer, P., Gruber, N., Bakker, D. C. E., & Schuster, U. (2014). Recent variability of the global ocean carbon sink. *Global Biogeochemical Cycles*, *28*(9), 927–949. https://doi.org/10.1002/2014GB004853

Landschützer, P., Gruber, N., Bakker, D. C. E., Stemmler, I., & Six, K. D. (2018). Strengthening seasonal marine $CO_2$ variations due to increasing atmospheric $CO_2$. *Nature Climate Change*, *8*(2), 146–150. https://doi.org/10.1038/s41558-017-0057-x

Lavoie, D., Lambert, N., Starr, M., Chassé, J., Riche, O., Le Clainche, Y., et al. (2021). The Gulf of St. Lawrence Biogeochemical Model: A Modelling Tool for Fisheries and Ocean Management. *Frontiers in Marine Science*, *8*, 732269. https://doi.org/10.3389/fmars.2021.732269

Roobaert, A., Regnier, P., Landschützer, P., & Laruelle, G. G. (2024). A novel sea surface pCO2-product for the global coastal ocean resolving trends over 1982–2020. *Earth System*

*Science Data*, *16*(1), 421–441. https://doi.org/10.5194/essd-16-421-2024

Rutherford, K., Fennel, K., Atamanchuk, D., Wallace, D., & Thomas, H. (2021). A modelling study of temporal and spatial $pCO_2$ variability on the biologically active and temperature-dominated Scotian Shelf. *Biogeosciences*, *18*(23), 6271–6286. https://doi.org/10.5194/bg-18-6271-2021

Signorini, S. R., Mannino, A., Najjar, R. G., Friedrichs, M. A. M., Cai, W.-J., Salisbury, J., et al. (2013). Surface ocean *p*$CO_2$ seasonality and sea-air $CO_2$ flux estimates for the North American east coast. *Journal of Geophysical Research: Oceans*, *118*(10), 5439–5460. https://doi.org/10.1002/jgrc.20369

Vandemark, D., Salisbury, J. E., Hunt, C. W., Shellito, S. M., Irish, J. D., McGillis, W. R., et al. (2011). Temporal and spatial dynamics of $CO_2$ air-sea flux in the Gulf of Maine. *Journal of Geophysical Research*, *116*(C1), C01012. https://doi.org/10.1029/2010JC006408

Wu, Z., Wang, H., Liao, E., Hu, C., Edwing, K., Yan, X.-H., & Cai, W.-J. (2024). Air-sea CO2 flux in the Gulf of Mexico from observations and multiple machine-learning data products. *Progress in Oceanography*, *223*, 103244. https://doi.org/10.1016/j.pocean.2024.103244